# Oxytocin pathway gene networks in the human brain

Daniel S. Quintana [1], Jaroslav Rokicki[1,2], Dennis van der Meer[1], Dag Alnæs[1], Tobias Kaufmann[1], Aldo Córdova-Palomera[1], Ingrid Dieset[1], Ole A. Andreassen [1] & Lars T. Westlye [1,2]

Oxytocin is a neuropeptide involved in animal and human reproductive and social behavior. Three oxytocin signaling genes have been frequently implicated in human social behavior: *OXT* (structural gene for oxytocin), *OXTR* (oxytocin receptor), and *CD38* (oxytocin secretion). Here, we characterized the distribution of *OXT*, *OXTR*, and *CD38* mRNA across the human brain by creating voxel-by-voxel volumetric expression maps, and identified putative gene pathway interactions by comparing gene expression patterns across 20,737 genes. Expression of the three selected oxytocin pathway genes was enriched in subcortical and olfactory regions and there was high co-expression with several dopaminergic and muscarinic acetylcholine genes, reflecting an anatomical basis for critical gene pathway interactions. fMRI meta-analysis revealed that the oxytocin pathway gene maps correspond with the processing of anticipatory, appetitive, and aversive cognitive states. The oxytocin signaling system may interact with dopaminergic and muscarinic acetylcholine signaling to modulate cognitive state processes involved in complex human behaviors.

[1] NORMENT, KG Jebsen Centre for Psychosis Research, Division of Mental Health and Addiction, University of Oslo, and Oslo University Hospital, PO Box 4956 Oslo, Norway. [2] Department of Psychology, University of Oslo, Oslo 0373, Norway. These authors contributed equally: Daniel S. Quintana, Jaroslav Rokicki. Correspondence and requests for materials should be addressed to D.S.Q. (email: daniel.quintana@medisin.uio.no)

Oxytocin is an evolutionarily conserved neuropeptide implicated in an array of social and reproductive behaviors; its role in complex behavioral traits and in the pathophysiology of mental health conditions has attracted considerable attention[1]. Oxytocin is mostly synthesized in neurons in the supraoptic nucleus and the paraventricular nucleus for both systemic and central release. Research on humans has shown beneficial effects of intranasal oxytocin on performance on tests assessing social cognition[2] and gaze to the eye region[3]. Moreover, single nucleotide polymorphisms in oxytocin pathway genes have been associated with social behavior and psychiatric disorders[4]. Emerging evidence also points to the oxytocin system's role in energy metabolism, with relevance for metabolic functions[5].

The distribution of oxytocin receptor (OXTR) mRNA across the brain provides a proxy for the distribution of oxytocin binding[6], allowing for a detailed mapping of the anatomical geography of the oxytocin system in the brain. Seminal animal work using histochemistry and immunohistochemistry revealed high concentrations of OXTR mRNA in the hypothalamus, amygdala, olfactory bulb, ventral pallidum, and the dorsal vagal nucleus in rodents[7,8]. Further, experimentally increasing[9] or decreasing[10] OXTR expression in the prairie vole nucleus accumbens modulated partner preference behavior, suggesting a correspondence between the spatial distribution of OXTR mRNA, its functional neuroanatomy, and behavioral relevance. While OXTR mRNA localization in the rodent brain is well-described[11], its anatomical distribution across the human brain is poorly understood, as investigations have historically tended to sample very few brain regions[12]. While more recent work has exclusively examined whole-brain distribution of OXTR genes against a limited set of genes[13] and overexpression in broad functional brain networks (e.g., limbic network)[14], researchers have yet to explore OXTR's associations with all protein-coding genes or with gene expression patterns associated with specific cognitive states.

Characterizing interactions of OXTR with other key elements of the oxytocin signaling pathway and biological systems beyond this pathway is critical for determining oxytocin's behavioral and functional relevance. Along with OXTR, CD38 and OXT genes in the oxytocin signaling pathway have been implicated in human social behavior[4]. Specifically, CD38 is involved in oxytocin secretion via $Ca^{2+}$ release[15], and oxytocin-neurophysin I (OXT) encodes the oxytocin prepropeptide containing the nonapeptide oxytocin and the carrier protein neurophysin-I[16]. OXT mRNA has been shown to be highly expressed in human paraventricular nucleus of the hypothalamus, the lateral hypothalamic area, and the supraoptic nucleus, and there is evidence of co-expression with OXTR mRNA and the μ and κ types opioid receptor mRNA[13], providing a putative avenue for interactions between the oxytocin and opioid pathways. The interactions with the oxytocin system extend beyond the opioid pathway, including the dopaminergic[17,18] and muscarinic acetylcholine[12,19] circuits, with possible implications for social behavior and psychiatric disorders. For instance, the dopamine D2-receptor subtype (DRD2), has been implicated in various putative intermediate phenotypes in psychiatric illness, including motivational processing[20] and pair bonding in animal models[21]. Moreover, the muscarinic acetylcholine M4 receptor (CHRM4) has been associated with schizophrenia[22] and implicated in cognitive flexibility[23] and dopamine release[24]. Finally, oxytocin can also bind to AVPR1A receptors[25], which have also been linked to social functioning[26]. However, the mRNA co-expression of these systems, and potentially others, with the oxytocin system is not well characterized.

While brain regions with high oxytocin pathway gene expression in humans have been identified[12,13], inference of specific cognitive states from the activity of single brain regions is elusive.

For instance, commonly observed increases in medial and lateral frontal region activity during emotion and pain processing seem to be better explained by more general sustained attention processes[27]. By collecting data from more than 14,000 fMRI studies, NeuroSynth[27] allows for reverse inference of cognitive states based on a given brain gene expression map with high specificity. Establishing the specific cognitive state correlates of oxytocin pathway genes will provide a deeper understanding of the human oxytocin system and its relevance for brain functions and mental health.

Here we characterize the anatomical distribution of oxytocin pathway mRNA expression in the human brain, identify putative gene interactions, and explore the functional relevance of these patterns. Expression of critical oxytocin pathway genes are enriched in subcortical and olfactory regions and these genes are highly co-expressed with several dopaminergic and muscarinic acetylcholine genes, along with genes associated with metabolic regulation. Moreover, OXTR and CD38 contribute to a co-expression gene network that is significantly enriched in a waist-hip ratio GWAS along with genesets associated with cognition and feeding behavior. The results from a large-scale fMRI meta-analysis suggest that oxytocin pathway gene expression maps correspond with the processing of anticipatory, appetitive, and aversive cognitive states. These results indicate that the oxytocin signaling system may operate synergistically with the dopaminergic and muscarinic acetylcholine signaling systems to exert its complex effects on cognition.

## Results

**Oxytocin gene expression patterns in the brain.** The full dataset of protein coding genes (n = 20,737) from six donor brains were collected from the Allen Human Brain Atlas (http://human.brain-map.org/). Each brain was sampled in 363–946 distinct locations, either in the left hemisphere only (n = 6), or over both hemispheres (n = 2) using a custom Agilent 8 × 60 K cDNA array chip. Analyses were performed on left hemisphere samples due to a larger sample size. Individual brain maps were non-linearly registered to the MNI152 (Montreal Neurological Institute) template using Advanced Normalization Tools[28]. Next, we extracted region specific statistics for 54 brain regions based on the Automated Anatomical Label (AAL) atlas.

Of primary interest were the three oxytocin pathway genes that have been associated with social behavior in both animal and human research: OXTR, CD38, and OXT[4]. Four other selected sets of mRNA, which are thought to co-express with oxytocin pathway mRNA and modulate social behavior, were also of specific interest: A dopaminergic set (DRD1, DRD2, DRD3, DRD4, DRD5, COMT, and DAT1)[17,18], a muscarinic acetylcholine set (CRHM1, CRHM2, CRHM3, CRHM4, and CRHM5)[12,19], a vasopressin set (AVPR1A, AVPR1B)[26,29] and an opioid set (OPRM1, OPRD1, and OPRK1)[13]. Of additional interest were complete sets of dopaminergic (n = 63), cholinergic (n = 79), and oxytocinergic (n = 94) genes, which were identified using the Kyoto Encyclopaedia of Genes and Genomes (KEGG) database (http://www.kegg.jp). Full gene set lists are provided in Supplementary Data 1.

The Genotype-Tissue Expression (GTEx) project database[30] was used for independent sample validation. Although this dataset provides gene expression data from fewer brain regions (i.e., 10) compared to the Allen dataset, the data is derived from a larger dataset of donors (mean sample size for mRNA expression across brain regions = 131.7, range = 88–173). Median gene expression profiles from 10 distinct brain regions were extracted for the above-specified 20 genes of interest from the GTEx database and median values were calculated for these same 10

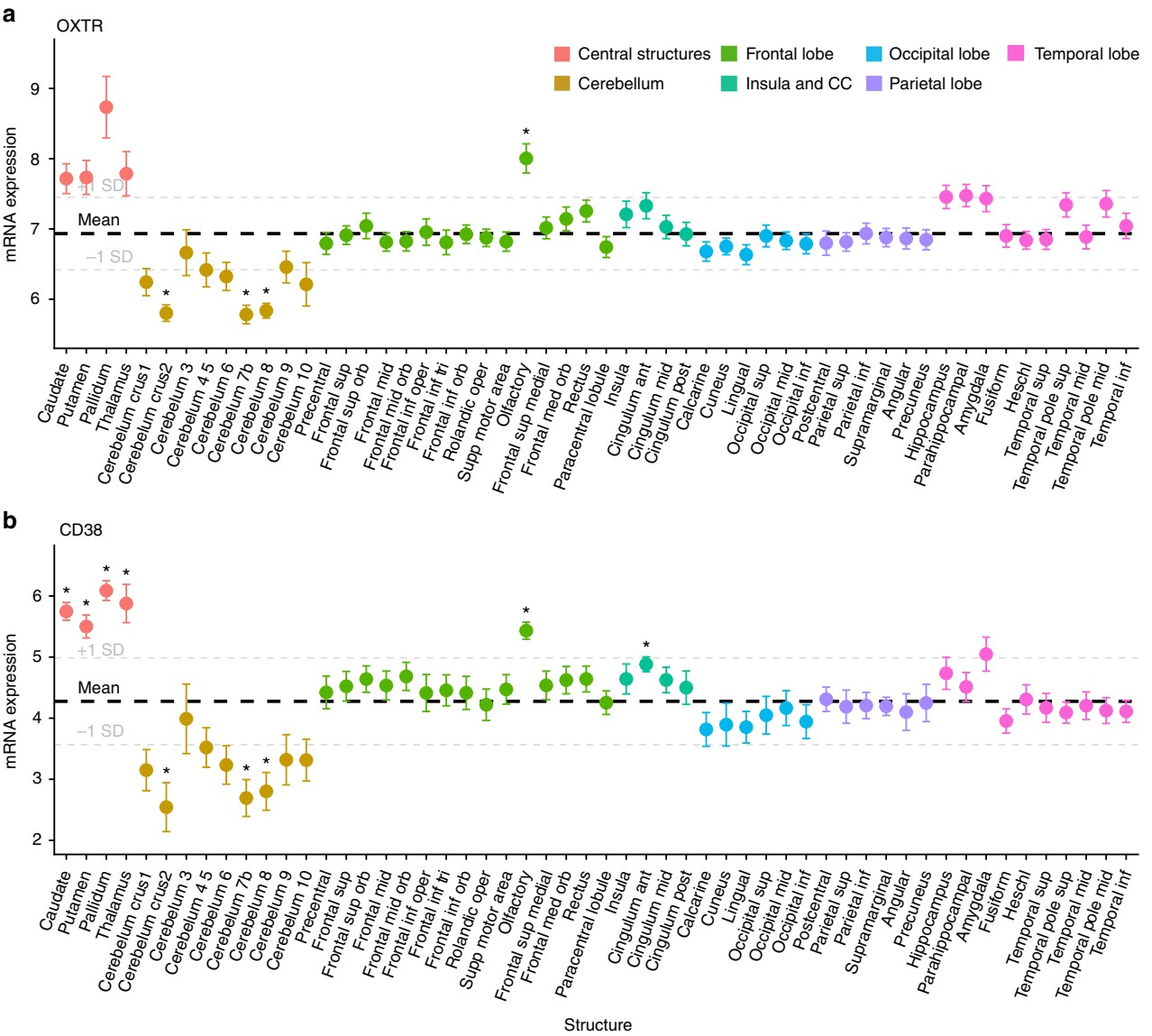

**Fig. 1** Oxytocin pathway gene expression in the human brain. Each point represents mean expression from six donors with standard errors for a given brain region for **a** OXTR and **b** CD38. The bolded dashed lines represent mean expression across the all regions with 1 standard deviation (+/−) also shown. Compared to average brain expression, there is increased expression of *OXTR* and *CD38* in central and temporal brain structures, along with the olfactory region. Lower than average expression is observed in the cerebellum. *$p < 0.05$ (FDR corrected for 54 tests)

regions from the Allen dataset (see Methods, Supplementary Fig. 1, and Supplementary Data 1). For independent sample validation, the rank-order correlation of gene expression between the Allen and GTEx datasets using the 10 brain regions reported in the GTEx database was calculated.

**Region-of-interest gene expression**. Compared to average expression across the brain, statistically significant higher expression of *OXTR* mRNA levels was observed in the olfactory bulbs ($p = 0.048$, $d = 2.1$; Supplementary Data 1). Subcortical structures had higher than average expression of *OXTR*, associated with Cohen's $d$ values >1, but these did not reach FDR corrected (54 tests) thresholds of statistical significance (Pallidum: $p = 0.1$, $d = 1.7$; Caudate: $p = 0.12$, $d = 1.5$; Putamen: $p = 0.13$, $d = 1.3$; Thalamus, $p = 0.18$, $d = 1.1$). There was significantly lower *OXTR* expression in the cerebral crus II ($p = 0.01$, $d = 4$), cerebellum 7b ($p = 0.01$, $d = 3.6$), and cerebellum 8 ($p = 0.01$, $d = 4.3$; Fig. 1a). *OXTR* expression patterns in the brain were comparable to patterns in the GTEx database ($r_s = 0.81$, $p <$

0.01). *CD38* gene expression was statistically significantly higher than average in the caudate ($p = 0.004$, $d = 4.1$), pallidum ($p = 0.004$, $d = 4.6$), olfactory bulbs ($p = 0.01$, $d = 3.4$), putamen ($p = 0.02$, $d = 2.7$), thalamus ($p = 0.03$, $d = 2.1$), and cingulate anterior ($p = 0.03$, $d = 2$; Fig. 1b). There was lower *CD38* expression in the cerebral crus II ($p = 0.045$, $d = 1.8$), cerebellum 7b ($p = 0.03$, $d = 2.1$), and cerebellum 8 ($p = 0.03$, $d = 1.9$). *CD38* expression patterns in the Allen database were comparable to *CD38* expression patterns in the GTEx database ($r_s = 0.67$, $p = 0.04$). There were no brain regions with significantly higher than average OXT expression after FDR correction (54 tests; Supplementary Fig. 2). *OXT* expression patterns in the Allen database were not significantly associated to the GTEx expression patterns ($r_s = 0.22$, $p = 0.54$). Expression of genes included in the selected dopaminergic, muscarinic acetylcholine, vasopressin, and opioid sets are presented in Supplementary Figures 3–6, respectively. Of note, *DRD1*, *DRD2*, *CHRM4*, and *OPRK1* genes were highly expressed in subcortical regions, including the caudate, pallidum, putamen, and thalamus (Supplementary Data 1).

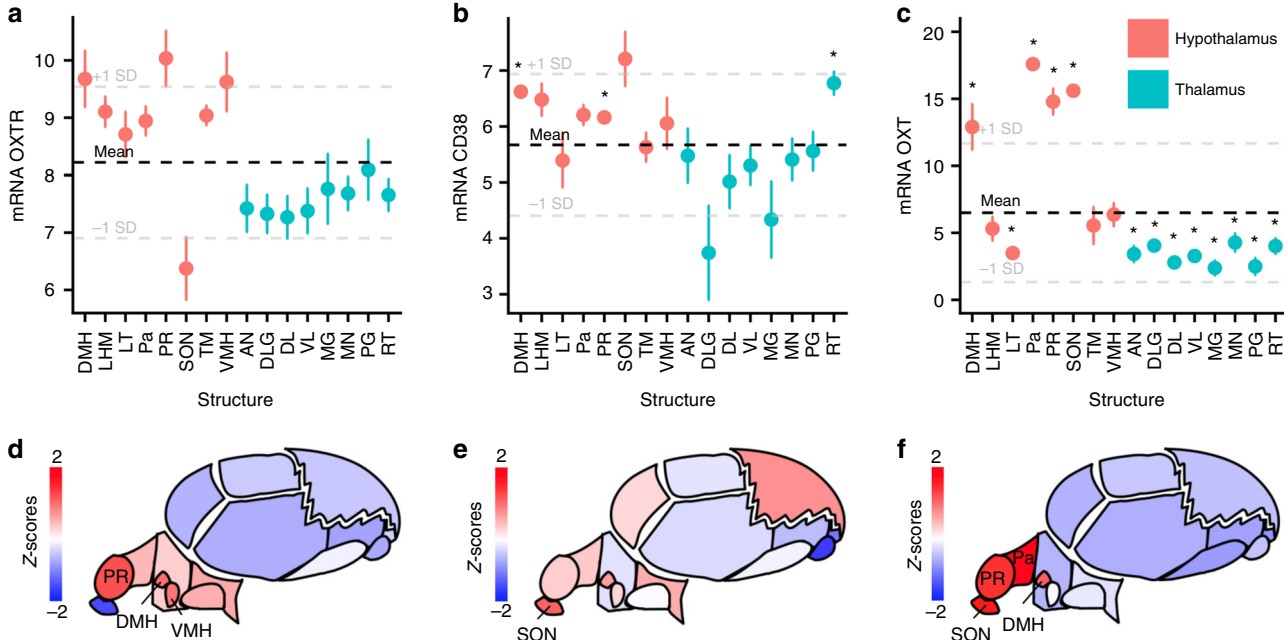

**Fig. 2** Expression of *OXTR*, *CD38*, and *OXT* in hypothalamic and thalamic substructures. Each point represents mean expression of **a** *OXTR*, **b** *CD38*, and **c** *OXT* with standard errors for a given brain region. The bolded dashed lines represent mean expression across the all regions with 1 standard deviation (+/−) also shown. *Z*-scores for **d** *OXTR*, **e** *CD38*, and **f** *OXT* expression are also presented in anatomic maps for hypothalamic and thalamic substructures (see Supplementary Fig. 7 for map legend). Substructures with expression more than 1 standard deviation greater than the mean are labeled. DMH: dorsomedial hypothalamic nucleus, LMH: lateral hypothalamic area (mammillary region), LT: lateral tuberal nucleus, Pa: paraventricular nucleus of the hypothalamus, PR: Preoptic region, SON: supraoptic nucleus, TM: tuberomammillary nucleus, VMH: ventromedial hypothalamic nucleus, AN: anterior group of nuclei, DLG: dorsal lateral geniculate nucleus, DL: lateral group of nuclei (dorsal division), VL: lateral group of nuclei (ventral division), MG: medial geniculate complex, MN: medial group of nuclei, PG: posterior group of nuclei, RT: reticular nucleus of thalamus. * *p* < 0.05 (FDR corrected for 16 tests)

Given the role of the hypothalamus in oxytocin signaling, expression patterns within hypothalamic substructures were also summarized using anatomic labels from the Allen dataset (Fig. 2; Supplementary Data 1). Oxytocin pathway gene expression in nearby thalamic substructures using anatomic labels from the Allen dataset were also summarized for comparison (see Supplementary Fig. 7 for anatomical map of substructures). There was significantly higher expression in the hypothalamic structures compared to the thalamic structures for *OXTR* (*p* < 0.001), *CD38* (*p* = 0.002) and *OXT* (*p* < 0.001). Notably, there was significantly higher expression of *OXT* mRNA in the PVN, SON, preoptic region, and dorsomedial hypothalamic nucleus substructures compared to mean expression across the hypothalamus and thalamus (FDR corrected *p* < 0.001).

**Voxel-by-voxel gene expression associations.** Using the AAL atlas is a straightforward means to categorize and identify brain regions associated with increased gene expression. However, investigating specific brain regions of interest reduces spatial resolution. This is particularly an issue when correlating the expression patterns of two genes over a set of brain regions, as smaller anatomical regions may bias overall associations. Moreover, results may differ according to the selected atlas. Therefore, to increase precision, reduce the risk of bias, and examine associations independent of atlas, we used voxel-by-voxel mRNA maps for analysis (see Methods).

A Spearman's correlation matrix demonstrating the relationships between genes of interest using the raw Allen data is presented in Fig. 3a. Hierarchical clustering revealed that oxytocin pathway genes (*OXTR*, *CD38*) cluster with some elements of the dopaminergic (*DRD2*, *DRD5*, and *COMT*) and cholinergic (*CHRM4*, *CHRM5*) systems, but not all, indicating the

relationship between oxytocin pathway genes and dopaminergic/cholinergic gene sets were not globally strong. There were statistically significant relationships between the expression of the genes in this cluster (all FDR adjusted *p*-values < 0.05). The correlation of *OXTR*, *CD38*, and *OXT* within complete sets of dopaminergic, cholinergic, and oxytocinergic genes are visualized in Fig. 3b. For an independent sample validation of the clustering pattern of these 20 specified genes of interest, gene co-expression in the Allen dataset was compared with the independent GTEx dataset (see Methods). In both datasets, *OXTR*, *CD38*, *DRD2*, and *CHRM4* clustered together (Supplementary Fig. 8). To explore whether these co-expression patterns are not driven by gene expression differences between cortical and non-cortical areas, we visualized the similarities in expression patterns based on cortical areas only for comparison against whole brain patterns (Fig. 3c). A comparison of expression patterns across all the brain regions with only cortical samples suggest that *OXTR* and *CD38* co-expression patterns appear to be global, whereas *OXT* co-expression seems to be driven by correlations within cortical areas. The differential expression of the geneset cluster including *OXTR* and *CD38* (Fig. 3a) in 30 tissue types across the body was assessed using FUMA, which extracted data from the GTEx database (version 7)[30,31]. Hypergeometric tests were used to assess statistically significant enrichment, revealing that the *OXTR*/*CD38* geneset cluster was significantly enriched in brain tissue (Bonferroni corrected *p*-value < 0.05; Fig. 3d).

To uncover co-expression patterns of the full oxytocin signaling pathway (94 genes), we applied weighted gene co-expression network analysis using the WGCNA R package[32]. This analysis identified a co-expression module containing both *OXTR* and *CD38*, among 28 genes in total (Fig. 4a, Supplementary Data 1; Supplementary Fig. 9). Submission of the *OXTR*/*CD38* module to FUMA revealed that this geneset module is enriched in

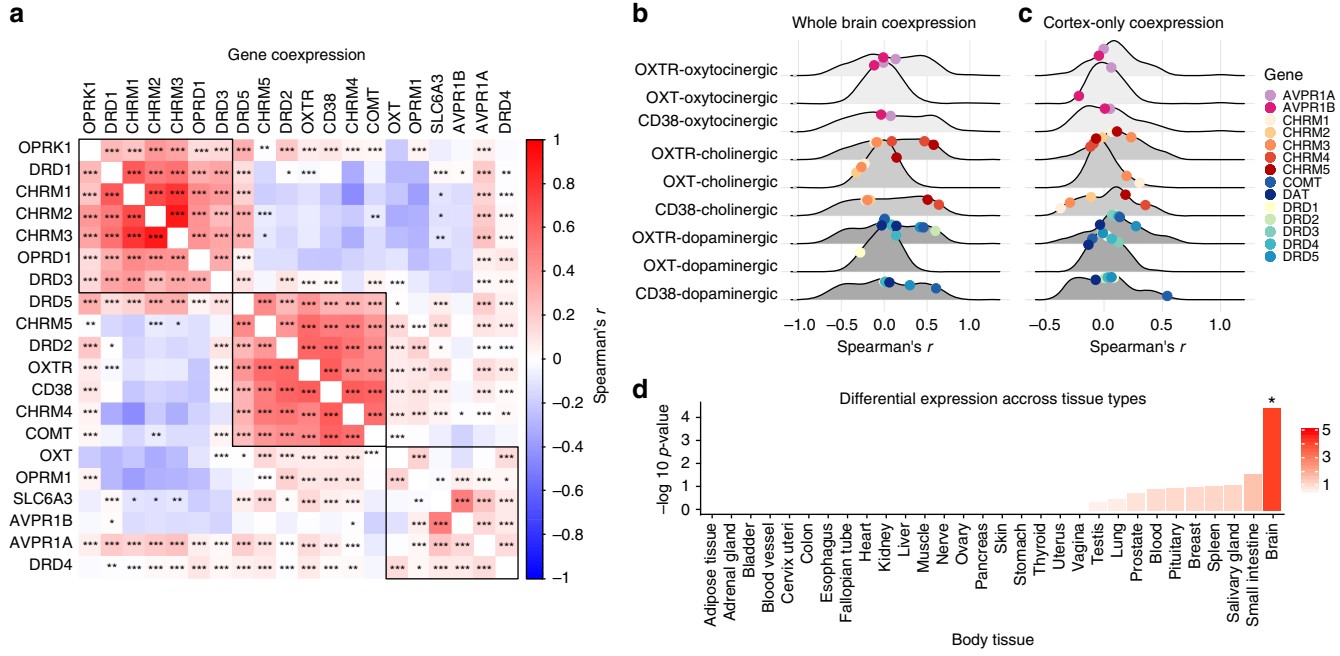

**Fig. 3** Co-expression of selected oxytocin, dopaminergic, muscarinic acetylcholine, vasopressin, and opioid gene sets in the brain. **a** Co-expression patterning for the expression of selected oxytocin, dopaminergic, muscarinic acetylcholine, vasopressin, and opioid gene sets. The complete linkage method was used to identify 3 clustering groups (black squares). *OXTR* and *CD38* are clustered together, along with *DRD2, DRD5, COMT, CHRM4, and CHRM5*. **b** Whole-brain co-expression of *OXTR*, *CD38*, and *OXT* with selected vasopressin, muscarinic acetylcholine, and dopaminergic gene sets. Spearman's correlations are visualized on a density distribution demonstrating all co-expression correlations between the of *OXTR*, *CD38*, and *OXT* and all genes in the complete oxytocinergic ($n = 94$), cholinergic ($n = 79$), and dopaminergic ($n = 63$) gene sets (see Supplementary Data 1 for full lists). **c** Cortex-only co-expression of *OXTR*, *CD38*, and *OXT* with selected vasopressin, muscarinic acetylcholine, and dopaminergic gene sets. **d** Differential expression of the OXTR/CD38 gene sets in 30 tissue types from the GTEx dataset, with -log 10 *p*-values representing the probability of the hypergeometric test. ***$p <$ 0.001, **$p <$ 0.01, *$p <$ 0.05

GWAS catalog reported genes[33] for waist-hip ratio adjusted for BMI and smoking ($p = 1.13 \times 10^{-2}$). This *OXTR/CD38* module was also enriched in genesets reported in the Molecular Signatures Database[34] that are associated with several behavioral and cognitive state processes including feeding behavior ($p = 5.83 \times 10^{-3}$), cognition ($p = 7.44 \times 10^{-3}$), and behavioral response to stimuli ($p = 1.05 \times 10^{-2}$). This *OXTR/CD38* module was also enriched in brain and breast tissue (Bonferroni corrected *p*-value < 0.05; Fig. 4b). Altogether, these results suggest that the identified *OXTR/CD38* module is biological meaningful. The association between *OXTR* and *CD38* was statistically significant ($p < 0.0001$) and among the top 5% of correlations compared to all 20,737 protein coding genes. Density plots of the correlations between the three oxytocin pathway mRNA probes and all other available protein coding genes are presented in Supplementary Figure 10.

Correlation analyses using Spearman's coefficient with all available 20,737 protein coding gene probes in the Allen Human Brain Atlas and each of the three oxytocin pathway gene probes are summarized in Supplementary Table 1, with the top 10 strongest positive and top 10 strongest negative correlations (the full set of correlations with the remaining 20,736 genes are presented in Supplementary Data 2). The *p*-values for these correlations were all < 0.001. Among the top ten most positive mRNA map correlations for *OXTR* were Neurotensin Receptor 2 (*NTSR2*; $r_s = 0.78$), Glutamate dehydrogenase 2 (*GLUD2*; $r_s = 0.78$), and Glutamate dehydrogenase 1 (*GLUD1*; $r_s = 0.77$), and the top ten most positive correlations with *CD38* included *NTSR2*

($r_s = 0.82$), *C12orf39* (Spexin; $r_s = 0.81$), and *GLUD1* ($r_s = 0.78$). Associations with these specific genes are notable given their reported role in metabolic regulation[35–38]. Additionally, the gene with the 6th strongest association with *CD38* was *PSAT1* ($r_s = 0.79$), whose disruption has been linked to schizophrenia[39].

**Donor-to-donor reproducibility of gene expression patterns**. Given differences in sex and ethnicity among donors, we examined the similarity of gene expression patterns across the six brains in the sample by using the concept of differential stability, which is the average Spearman's correlation between any possible combination of the 15 pairs between the six donors. This approach has previously been applied to the same dataset[40], with data suggesting that genes with strong differential stability are highly biologically relevant. We found that both *OXTR* and *CD38* had differential stability values in the top decile of all 20,737 protein coding genes in the dataset (Fig. 5a), indicating that gene expression patterning is reproducible, regardless of individual differences, such as ethnicity and sex. We also calculated donor-to-donor associations (Spearman's rank correlation coefficient) of gene expression patterns, revealing statistically significant associations ($p < 0.0001$) for *OXTR*, *CD38*, and *OXT* expression patterns between all donors (Fig. 5b–d). Inspection of gene expression data from the GTEx database also corroborates our differential stability data as this did not reveal sex differences in oxytocin pathway gene expression in the brain (Supplementary Fig. 11).

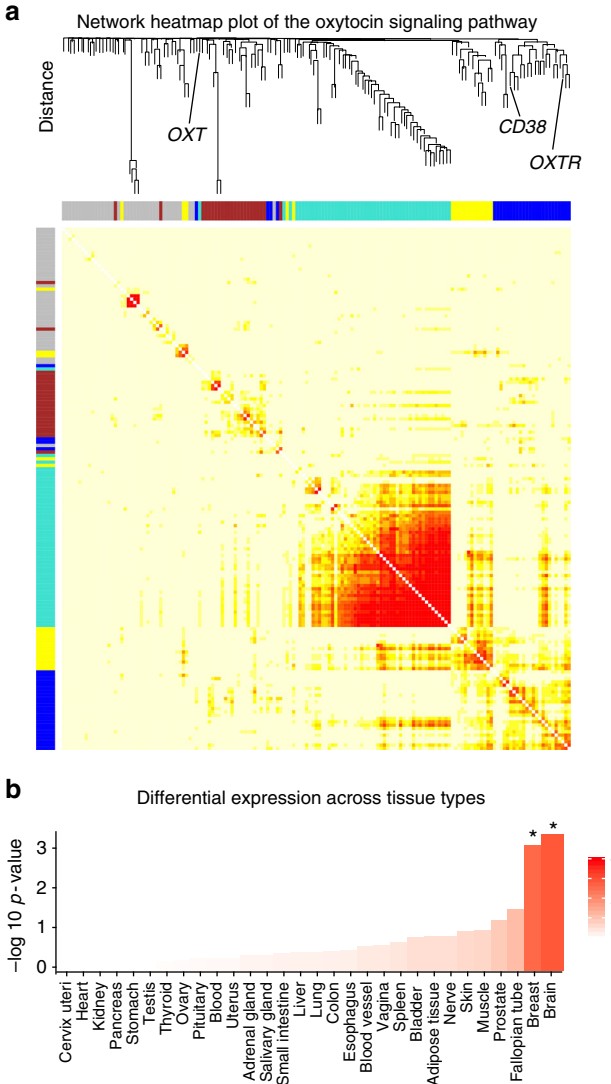

**Fig. 4** Co-expression patterns of the full oxytocin signaling pathway. **a** Weighted gene co-expression network analysis was used to construct a hierarchical clustering tree (also see Supplementary Fig. 9) and network heatmap plot of gene–gene connectivity of full oxytocin signaling pathway in the brain. Darker red colors represent stronger topological overlap. This analysis identified a module (dark blue) containing both *OXTR* and *CD38* (see Supplementary Fig. 9 and Supplementary Data 1 for a full list of genes and modules). **b** Differential expression of the OXTR/CD38 module in 30 tissue types from the GTEx dataset, with -log 10 *p*-values representing the probability of the hypergeometric test. *$p < 0.05$

**Cognitive state term correlates**. The NeuroSynth framework[27] has extracted text and neuroimaging data from 14,371 fMRI studies (release 0.7). While this tool can be used to create meta-analytic brain activation maps for specific cognitive states (e.g., stress) using forward inference, it can also be used to "decode" cognitive states, given an activation map, via reverse inference. Here, the term "cognitive state" refers to the neural processing that occurs response to a specific stimulus (e.g., pain) or that underlies a psychological process (e.g., learning).

To identify cognitive state correlates using brain maps of gene expression[41] we performed quantitative reverse inference via large-scale meta-analysis of functional neuroimaging data using mRNA brain expression maps on voxel-by-voxel left hemisphere brain maps, representing the average of 6 individuals (see

Methods). Using the NeuroSynth framework, we correlated our voxel-by-voxel mRNA maps (Fig. 6) with association *Z* maps, reflecting the posterior probability maps, which display the likelihood of a given cognitive state term being used in a study if activation is observed at a particular voxel. We assessed the 15 strongest relationships (Spearman's *r*) between *OXTR*, *CD38*, and *OXT* expression maps (5 unique terms for each gene) and all available cognitive state maps available in the NeuroSynth database. To assess the comparative strength of these relationships compared to other genes, we also assessed the relationship between these 15 cognitive state maps and the 20,737 mRNA gene maps. We plotted these distributions and identified the rank of *OXTR*, *CD38*, and *OXT* Spearman's *r* values compared to all other associations.

Decoding cognitive states meta-analytically from voxel-by-voxel mRNA maps (Fig. 6) via quantitative reverse inference revealed that *OXTR*, *CD38, and OXT* mRNA expression maps were most highly correlated with functional imaging maps that can be broadly categorized as anticipatory, appetitive, and aversive (Fig. 7a, Supplementary Table 2). The "Emotional" and "Facial expression" cognitive states were considered both aversive and appetitive given that emotions and facial expressions can either be positive or negative and that intranasal oxytocin has been shown to modulate the processing of both positive and negative emotions[42]. Figure 7b shows the full distribution of Spearman's correlation coefficients for each cognitive state term across all 20,737 gene maps, with labeled oxytocin pathway genes and their absolute rank for each term (Supplementary Table 2). Notably, the *OXTR* map's relationship with the cognitive state maps of "sexual", "motivation", "incentive", and "anxiety" were all ranked among the top 0.5% strongest associations out of 20,737 genes and were statistically significant ($p < 0.001$; Fig. 7b; Supplementary Table 2), The "taste", "stress", "reward", "monetary", "fear", and "emotional" cognitive state maps were within the top 2.5% of all associations with *OXTR* expression and also statistically significant ($p < 0.001$; Fig. 7b; Supplementary Table 2). In other words, not only were these cognitive state maps the most highly correlated with the *OXTR* expression map, but these correlations with specific cognitive states were amongst the strongest correlations across all 20,737 protein coding genes. All correlation coefficient *p*-values for the relationship between each cognitive state term and oxytocin pathway genes (*OXTR, CD38, and OXT*) are presented in Supplementary Table 2. Additionally, *OXTR* ($p = 0.0002$; Fig. 7c) and *CD38* ($p = 0.006$; Fig. 7d), had significantly greater mRNA expression in brain regions associated with social behavior compared to brain regions associated with non-social cognitive states (Supplementary Data 1). There was no significant difference in *OXT* mRNA expression ($p = 0.14$; Fig. 7e) between these social and non-social brain regions.

## Discussion

The anatomical distribution of gene expression in the brain is heterogeneous and highly coordinated[40]. Dynamic alterations in gene expression are essential in response to environmental demands, and are critically involved in a range of cognitive states, learning processes, and diseases[43,44], while the basic organization likely partly reflects the evolutionary conserved modular layout of the brain[45]. Gene-gene co-expression patterns form specific genetic signatures in the brain, representing distinct tissues and biological systems[40], and likely reflect the potential of complex and differential gene-gene interactions with implications for brain disorders and cognitive health. Here, we leveraged the unique human brain mRNA expression library from Allen Brain Atlas to show that mRNA reflecting specific genes in the oxytocin pathway are highly expressed in subcortical and temporal brain

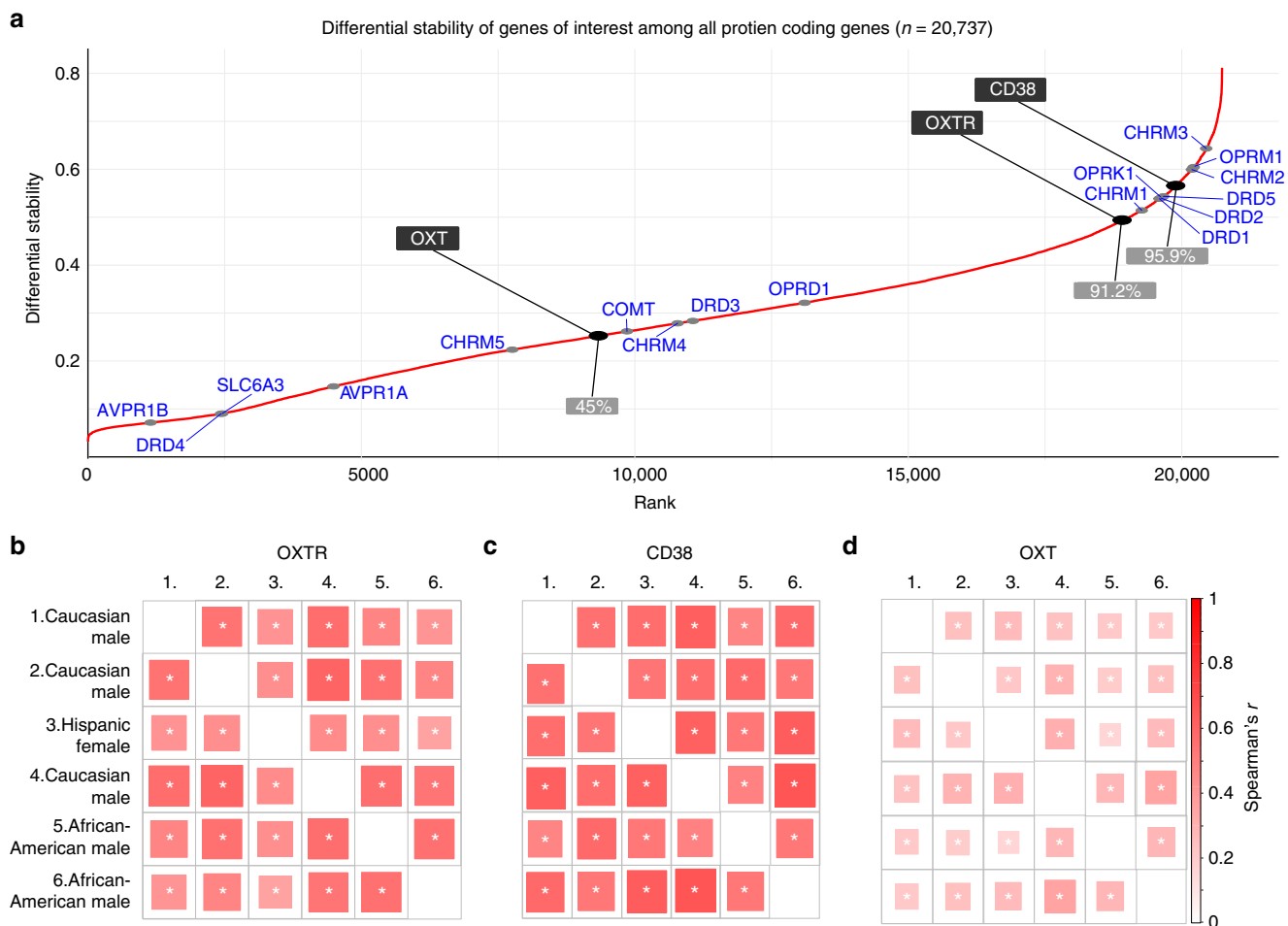

**Fig. 5** Differential stability of protein-coding genes. **a** Differential stability for all protein coding genes ($n = 20,737$) was calculated to assess the similarity of gene expression patterns from donor-to-donor. *OXTR* and *CD38* were in the top decile of all genes (ranking in gray boxes), suggesting high reproducibility. Donor-to-donor differential stability was also assessed for **b** *OXTR*, **c** *CD38*, and **d** *OXT*. Each individual association for all three genes were statistically significant, suggesting that gene expression patterns were stable between each donor (ethnicity and sex are presented for each donor). *$p < 0.001$

structures, along with the olfactory region. We also show reduced expression of *OXTR* and *CD38* in the cerebellum, consistent with prior animal research[46]. Importantly, an independent sample comparison revealed that the observed *OXTR* and *CD38* expression patterns from the Allen database were consistent with *OXTR* and *CD38* expression patterns observed in the GTEx database. Expression patterns of *OXT* in the Allen database were less stable from donor-to-donor and were not significantly related to the expression patterns in the GTEx database, which limits the generalizability of the results relating to this specific gene. *OXT* expression did not demonstrate strong whole-brain correspondence with *OXTR* and *CD38* expression, which was likely due to *OXT* transcripts only showing a strong presence in oxytocin producing cells within hypothalamic regions.

*OXTR* and *CD38* showed considerable co-expression with *DRD2*, *DRD5*, *COMT*, *CHRM4*, and *CHRM5* genes, providing evidence for putative interactions between dopaminergic and muscarinic acetylcholine systems with oxytocin pathway signaling. Exploratory analysis between oxytocin pathway mRNA and 20,737 mRNA probes revealed several relationships worth noting in the context of metabolic and feeding regulation, as well as psychiatric disorders. We discovered that *OXTR* is highly co-expressed with *NTSR2*, which has been found to regulate ethanol consumption in mice[35] and *GLUD1* and *GLUD2*, which are involved in energy homeostasis[36,37]. *CD38* is highly co-expressed

with *NTSR2* and *C12orf39* (Spexin), with the latter associated with weight regulation[38]. Moreover, weighted gene co-expression network analysis identified a co-expression module containing both *OXTR* and *CD38*, which was enriched in a waist-hip ratio GWAS[33] and a gene set associated with feeding behavior[34]. *OXTR* mRNA expression was also strongly associated with the neural processing of taste stimuli. Altogether, these results are consistent with emerging evidence that the oxytocin system plays a role in the metabolic and feeding dysregulation, which are overrepresented in severe mental illnesses[5]. Notably, research indicates that intranasal oxytocin administration improves appetite regulation in humans[47,48].

In regards to psychiatric disorders, there were strong negative correlations between oxytocin pathway genes and *PAK7* and *GABRD* which have been associated with schizophrenia[49–52]. *CD38* was highly correlated with *PSAT1*, a gene that has been associated with schizophrenia[39]. While these studies suggest intriguing links between specific genes and psychiatric illnesses, it is important to note that these are largely single report associations that require replication in independent samples. Quantitative reverse inference via meta-analysis of 14,371 fMRI studies revealed that the distribution of the oxytocin signaling pathway genes is most strongly associated with anticipatory, appetitive, and aversive cognitive states (Supplementary Table 2). Notably, out of the 20,737 gene expression brain maps that we compiled,

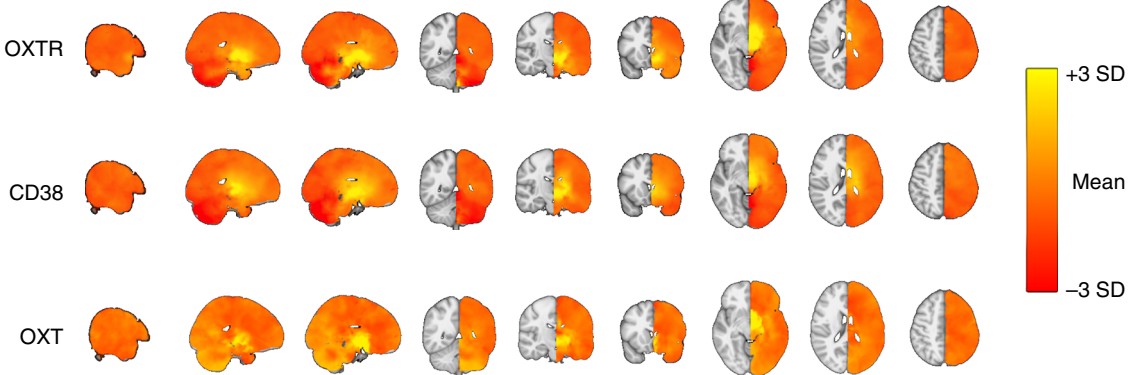

**Fig. 6** Voxel-by-voxel brain gene expression maps for *OXTR*, *CD38*, and *OXT*. mRNA expression maps were created using left hemisphere data from the Allen Human Brain Atlas dataset. These maps were submitted to the NeuroSynth framework to assess associations with cognitive state activation maps. SD: standard deviation

the *OXTR* expression map was among the top 0.5% of strongest relationships with the "sexual", "motivation", "incentive", and "anxiety", cognitive state maps. Put another way, these cognitive state maps had a stronger correlation with *OXTR* expression patterns than at least 99.5% of all protein coding genes in the brain. These results are also consistent with a growing body of research indicating that intranasal oxytocin administration modulates emotion processing[2,3,53,54], suggesting that intranasally administered oxytocin may exert its effects by binding to oxytocin receptors located in regions associated with emotion processing. Indeed, compelling rodent evidence suggests that direct oxytocin central administration influences social behavior and cognition via action on oxytocin receptors[55]. In these animals, oxytocin receptors are located in regions that are crucial for social behavior and the processing of social cues[11].

Data from both our co-expression and meta-analytic analyses provides converging evidence that oxytocin may exert its effects synergistically with the dopaminergic network[17]. The oxytocin and dopaminergic systems have been shown to work in concert to promote rodent maternal behaviors[56], and central D2 receptors modulate social functioning in prairie voles[57]. There is also evidence of direct dopaminergic fibers to the supraoptic nucleus and the paraventricular nucleus[58], consistent with demonstrations that dopamine may stimulate oxytocin release in the hypothalamus[59]. This is also congruous with reported deficits in dopaminergic signaling in schizophrenia[60] and autism[61]. The muscarinic acetylcholine system may also work with the oxytocin system to boost attention[62] and to assist with the release of oxytocin[63]. Relatedly, the muscarinic acetylcholine system is thought to contribute to the cognitive dysfunction observed in schizophrenia[64]. Altogether, our findings corroborate emerging evidence for interactions between the oxytocin, acetylcholine[12], and dopaminergic signaling systems[17].

Our data revealed high expression of oxytocin pathway mRNA in the olfactory region, which is in line with animal research[65]. Indeed, social olfactory deficits in mice without the oxytocin gene[66] are rescued with injection of oxytocin into rat olfactory bulbs[67]. Increased mRNA in the human olfactory region may be vestigial, as olfaction is not as important for human conspecific identification compared to most other mammals due to species specialization, however, intranasal oxytocin has been shown to improve olfactory detection performance in schizophrenia[68], and intranasal oxytocin is thought to enter the brain via the olfactory bulbs[69].

There are some important limitations worth noting. First, given the nature of human brain tissue collection for mRNA studies, the sample size was small and the donor group was variable in

regards to age and ethnicity. These issues, along with individual differences in brain sizes and morphological features, may have introduced variability in sampling. However, there are several reasons to believe that donor-to-donor variability did not influence the outcomes of the study. The relatively small amount of measurement error for the mRNA probes is indicative of high precision and sufficient generalizability. We also independently validated general gene expression patterning in the larger GTEx dataset, however, it is worth noting that this comparison is somewhat limited due to fewer brain tissue sampling sites in the GTEx dataset compared to the Allen dataset. While there was only 1 female in the sample, which prohibited a comprehensive examination of sex differences in mRNA distribution from the Allen dataset, whole-brain expression patterns of *OXTR*, *CD38*, and *OXT* from the female donor was significantly related to the expression patterns of all the other male donors. Moreover, data from the GTEx dataset demonstrated little evidence of sex differences in central oxytocin pathway gene expression, consistent with preliminary human research[12]. However, it could be possible that sex differences in oxytocin pathway gene expression exist in brain regions not sampled in the GTEx dataset. Finally, we demonstrated that *OXTR* and *CD38* demonstrate high differential stability, which is indicative of highly consistent patterning across donor brains[40]. Indeed, high differential stability appears to be a hallmark of genes with considerable biological relevance[40]. Second, despite animal models demonstrating that *OXTR* expression levels directly modulate social behavior[9,10] and that increased *OXTR* gene expression is associated with increased oxytocin binding, research is yet to demonstrate whether the increased expression of *OXTR* (as measured by the presence of its transcript) in humans is associated with the activation of the receptor during specific cognitive states.

To translate oxytocin to the clinic, research needs to demonstrate engagement of drug targets, such as *OXTR* occupancy reflected by regional brain activity changes[70]. Without precise targets, it is unclear whether non-significant effects of intranasal oxytocin, beyond insufficient statistical power[71], are due to an inefficacious drug or misidentified drug targets. By identifying precise oxytocin pathway targets in the human brain and the cognitive state correlates of these oxytocin pathway gene distribution, our study may help facilitate efforts to better understand oxytocin treatment efficacy as it identifies targets for oxytocin receptor engagement, which can facilitate dose-ranging studies[69,70]. Analysis of data from the Allen Brain Atlas database provides a unique opportunity to map the central oxytocin system and explore co-expression with other systems. Altogether, these results provide a proof-of-principle demonstration of

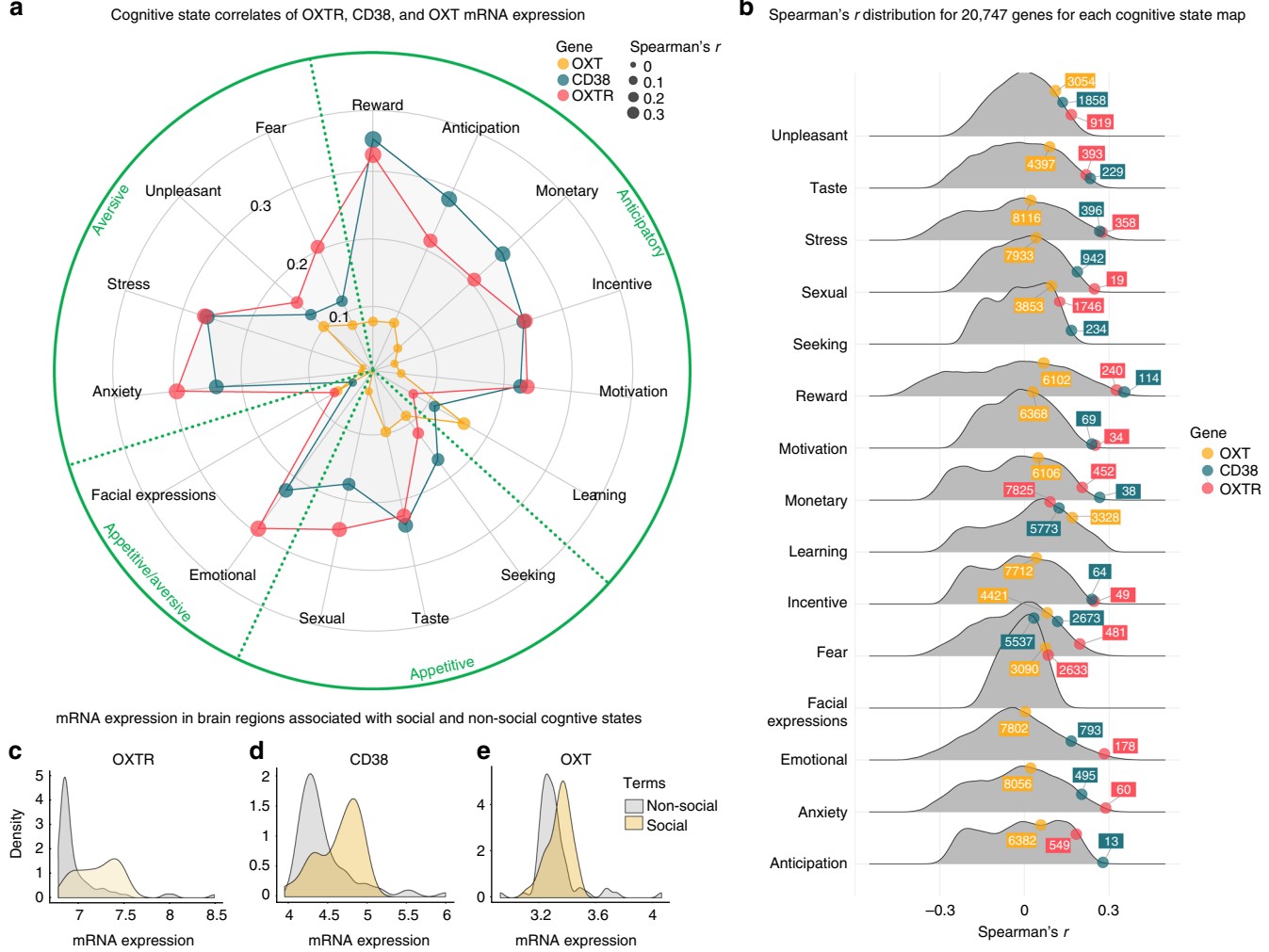

**Fig. 7** Cognitive state correlates of central oxytocin pathway gene expression patterns. **a** Cognitive states were meta-analytically decoded from central oxytocin pathway mRNA maps (Fig. 6) using the NeuroSynth framework. The top five strongest relationships for *OXTR*, *CD38*, and *OXT* (Spearman's *r*) are shown, with duplicates removed. **b** The distribution of Spearman's correlations between each protein coding gene map (*n* = 20,737) and cognitive state maps. The absolute ranking for each oxytocin pathway gene out of 20,737 correlations are shown (also see Supplementary Table 2). mRNA expression of **c** *OXTR*, **d** *CD38*, and **e** *OXT* in brain regions associated with social and non-social mental states (see Supplementary Data 1 for lists of social and non-social terms)

corresponding cognitive and gene expression patterns of a neuropeptide pathway involved in complex human behaviors.

## Methods

**Post-mortem brain samples**. mRNA distribution data was collected from the Allen Human Brain Atlas (http://human.brain-map.org/). Three donors were Caucasian males, one donor was a Hispanic female, and two donors were African-American males. Mean donor age was 42.5 (S.D. = 11.2) years. Data was collected 22.3 (S.D. = 4.5) hours after demise, on average (See Supplementary Table 3 for detailed donor profiles). If more than one probe for each mRNA was available, we selected the probe with the highest differential stability[40] which represented the probe with least amount of spatial variability among donors. The collection of human data complied with relevant ethical regulations, with institutional review board approval obtained at each tissue bank and repository that provided tissue samples. Moreover, informed consent was provided by each donor's next-of-kin. For more details regarding procedures and data collection associated with the Allen Human Brain atlas, see http://help.brain-map.org/display/humanbrain/Documentation.

**Independent sample validation**. Oxytocin pathway median expression values (Log10 transcripts per million) for 10 distinct brain regions (amygdala, anterior cingulate cortex, caudate, cerebellar hemisphere, cerebellum, cortex, frontal cortex, hippocampus, nucleus accumbens, putamen, substantia nigra) were extracted from the Genotype-Tissue Expression (GTEx) project database[30] for independent sample validation. Although this dataset offers less spatial resolution compared to the Allen dataset, the data is derived from a larger dataset of donors (mean sample size for mRNA expression across brain regions = 131.7, range = 88 to 173). For the comparison between the Allen and GTEx datasets, median gene expression for these 10 distinct GTEx regions were calculated from the Allen dataset (Supplementary Fig. 1), as the GTEx platform reports median values. Spearman's rank correlation coefficients ($r_s$) were calculated to assess the relationship between oxytocin pathway expression profiles from the two databases and overall co-expression patterns. Complete linkage clustering[72] was used to identify three clustering groups in both datasets.

**Voxel-by-voxel gene expression maps**. To create novel voxel-by-voxel volumetric expression maps, we first marked all the sample locations and expression values in native image space. To interpolate missing voxels, we labeled brain borders with the sample expression value that had the closest distance to a given border point (Supplementary Fig. 12). Next, we divided the space between scattered points into simplices based on Delaunay triangulation, then linearly interpolated each simplex with values to yield a completed map. All maps were computed in Matlab 2014a (The Mathworks Inc., Natick, MA, USA). We then created a composite brain map representing an average of 6 individuals on the left hemisphere, registered brains to MNI space using ANT's non-linear registration and averaged so that each gene's mRNA is represented by a single voxel-by-voxel brain map.

**Calculation of differential stability**. The average correlation (Spearman's $r$) across 15 pairs of 6 donors' voxel-by-voxel brain maps were used to calculate differential stability using an approach described by Hawrylytz and colleagues[40]. Since each voxel-by-voxel map was generated based on a limited and variable number of samples, to calculate the statistical significance of Spearman's coefficients we calculated $p$-values between two donors based on the smallest number of samples out of the two. That is, if one donor had samples from 353 locations and another one from 456, we would use the 353 samples to calculate a $p$-value for the pair.

**Statistical analysis of gene expression data**. The R statistical package (version 3.3.2) was used for statistical analysis. One-sample $t$-tests (two-tailed) were conducted to assess which of the 54 left hemisphere regions from six donor samples expressed mRNA to a significantly greater or lesser degree compared to average mRNA expression across the brain (for raw data, see Supplementary Data 1). To correct for multiple tests (54 in total), reported $p$-values were adjusted using a false discovery rate (FDR) threshold. Cohen's $d$ values for one-sample $t$-tests were calculated to yield a measure of effect size (Supplementary Data 1).

Based on raw expression data, we generated a $20 \times 20$ correlation matrix reflecting the spatial Spearman's correlation for each donor for each mRNA map pair including the following selected genes (Oxytocin pathway set: *OXTR*, *CD38*, *OXT*; Dopaminergic set: *DRD1*, *DRD2*, *DRD3*, *DRD4*, *DRD5*, *COMT*, and *DAT1*; muscarinic acetylcholine set: *CRHM1*, *CRHM2*, *CRHM3*, *CRHM4*, and *CRHM5*; vasopressin set: *AVPR1A*, *AVPR1B*; opioid set: *OPRM1*, *OPRD1*, *OPRK1*). Next, we computed the average correlation in each cell of the correlation matrix across the 6 donors and clustered the mRNA maps using the complete linkage method[72] to assess co-expression patterns. To assess putative gene–gene interactions beyond selected gene sets, we computed the spatial Spearman's correlations between all available protein coding mRNA probes ($n = 20{,}737$) and three oxytocin pathway mRNA probes. As more than one mRNA probe may have been available for a single gene, we selected the probe with the highest differential stability value. Unlike Pearson's $r$, which is the correlate statistic presented on the Allen Human Brain Atlas web interface, Spearman's correlation coefficient is less sensitive to outliers and non-normally distributed data.

To assess gene expression across various body tissues, normalized gene expression values (reads per kilo base per million; RPKM) were extracted from the GTEx database[30], via the FUMA platform[31]. As described by Watanabe and colleagues[31], normalized expression [zero mean of log2(RPKM + 1)] was used to assess differentially expressed gene sets. Bonferroni adjusted $p$-values are then calculated using two-sided $t$-tests per gene per tissue against all other tissues. Genes with a Bonferroni adjusted $p$-value $< 0.05$ and absolute log fold change $\geq 0.58$ were categorized as a differentially expressed gene set in a given tissue type. The presented -log10 $p$-values represent results from hypergeometric tests, which were used to assess if genes of interest were overrepresented in differentially expressed gene sets in specific tissues. Similarly, hypergeometric tests were used to assess if genes of interest were overrepresented in gene sets reported in the GWAS catalog[33] and gene sets associated with behavioral and cognitive state processes reported within GO biological processes gene sets within Molecular Signatures Database[34], using 20,119 protein coding genes as the background set. *P*-values were Benjamini-Hochberg adjusted for all genesets reported in the GO biological processes dataset and GWAS catalog, respectively.

**Weighted gene co-expression network analysis**. The WGCNA R package[32] was used to identify gene modules for expression of the oxytocin signaling pathway in the brain. A measure of topological overlap, which reflects biological meaningful models[73], was used to identify gene modules with tight interconnections. The WGCNA package uses average linked hierarchical clustering to group genes into modules based on their topological overlap. Soft thresholding power was selected based on a scale-free topology criterion. We selected power 10 (scale free topology $R^2 = 0.828$) as this was the lowest point for which the scale-free topology fit index curve flattens out (Supplementary Table 4).

**Cognitive state correlates using the NeuroSynth platform**. We correlated a voxel-by-voxel mRNA expression maps with NeuroSynth (version 0.3.7), a Python package for large-scale synthesis of functional neuroimaging data[27] to extract the top five cognitive correlates for each of three genes (*OXT*, *OXTR*, and *CD38*). We modified the NeuroSynth package to calculate Spearman's correlation coefficient instead of the default Pearson's correlation coefficient.

Next, to test specificity of these cognitive states (e.g. "Sexual", "Motivation"), we extracted association $Z$ maps, which reflect $Z$-scores of the association between the presence of activation and the presence of a feature. After extracting these maps, we calculated Spearman's correlation between all 20,737 genes and association $Z$ score maps and ranked them from largest to smallest.

A list of social and non-social cognitive states (Supplementary Data 1) was selected by two of the co-authors by consensus. Ambiguous cognitive states that could be easily classified as social or non-social (e.g., "Family") were excluded. Next, we extracted association $Z$ scores and FDR-corrected $p$-value maps for each cognitive state. We then thresholded each association map at FDR $p < 0.05$ and calculated the median association $Z$ score for each statistically significant region, which resulted in an expression score for each cognitive state term. Finally, we

calculated $p$-values by performing 5000 permutations for the difference in mean expression between social and non-social regions (Supplementary Data 1) via $t$-tests using the Permutation Analysis of Linear Models (PALM) tool[74].

## Data availability

Raw mRNA expression data is available from Allen Human Brain Atlas (http://human.brain-map.org) and GTEx (http://gtexportal.org). The Matlab script for producing the brain region-specific data, the resulting dataset, and the R script used for statistical analysis is available at https://osf.io/jp6zs/.

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

## Acknowledgements

This research was supported by an Excellence Grant from the Novo Nordisk Foundation to D.S.Q. (NNF16OC0019856), Research Council of Norway grants to O.A.A. (223273) and L.T.W. (249795), a KG Jebsen Stiftelsen grant to O.A.A. (SKGJ-MED-008), and South-Eastern Norway Regional Health Authority grants to O.A.A. (2017112) and L.T. W. (2014097 and 2015073).

## Author contributions

D.S.Q., J.R., D.v.d.M., and L.T.W. designed the study, analyzed the data, and interpreted the data. D.S.Q. and J.R. drafted the first version of the manuscript and D.v.d.M., D.A., T. K., A.C.-P., I.D., O.A.A., and L.T.W. revised it for important intellectual content.

## Additional information

**Competing interests:** The authors declare no competing interests.

