## [Peer Review File · Nature Communications]

Reviewers' comments:

Reviewer #1 (Remarks to the Author):

In the present manuscript the authors characterize the distribution of OXT, OXTR, and CD38 mRNA expression across the human brain and examine associations between gene transcription patterns and mental states through fMRI meta-analysis (NeuroSynth).

I found the paper to be of quite some interest and timely. However, I have several concerns/comments which I detail below.

1. Could the authors provide a rationale for why regions like the striatum are decomposed into sub-units (NAcc/caudate/putamen), but aggregate measures are given for regions like the thalamus and amygdala? It would be very useful to show this data given the heterogeneity of expression profiles across sub-structures.
2. Why is Figure 2B the same as Figure 3A? This redundancy should be removed.
3. "To our knowledge, research has yet to apply this approach to identify mental state correlates using brain maps of gene expression." Please refer to the bioarchive preprint by Fox et al. 2014 (Bridging psychology and genetics using large-scale spatial analysis of neuroimaging and neurogenetic data). Although not yet published, the work is a significant contribution and is parallel to the analyses in the current manuscript.
4. Could the authors specify which median GTEX values were used. Are these FPKM? RPKM?
5. On page 3 in the introduction, the authors suggest that "While central oxytocin receptor (OXTR) mRNA localization in the rodent brain is well-described, its anatomical distribution across the human brain is poorly understood, as investigations have tended to sample very few brain regions." I would refer them to the supplementary materials provided in the recently published NatComms paper by Anderson et al., 2018, which reports differential expression across large-scale brain networks, or the earlier work they cite from Steve Chang's lab (Dal Monte et al., 2017), which presents expression across a dense sampling of anatomical regions.

Reviewer #4 (Remarks to the Author):

This is an interesting manuscript that sheds light on the neural networks/gene pathways that mediate oxytocin's actions on human behavior. The manuscript also offers a strategy for understanding the actions of other brain-relevant genes on human behavior. Specific comments are as follows.

1. Abstract - the lines 25-29 that suggests why this study is potentially clinically relevant seems quite artificial and superficial. The article is of interest regardless of its supposed clinical implications. I urge the authors to delete these superfluous lines.
2. The concluding sentence in the Abstract - "These results provide a proof-of-principle demonstration of corresponding mental states with gene expression" - is quite weak and not very informative regarding the take home message for this manuscript. I strongly suggest revising the concluding statement.
3. I suggest deleting lines 62-64 (in line with my Comment 1).
4. The authors mention the opioid pathway "evidence of co-expression with 86 OXTR mRNA and the μ and κ types opioid receptor mRNA" and nevertheless don't examine its interactions with the

oxytocin system. Why?

5. The authors use the term "out-of-sample" validation. This term seems to have very specific definition regarding statistical model testing http://ec.europa.eu/eurostat/statistics-explained/index.php/Glossary:In-sample_vs._out-of-sample_forecasts

I wonder if this is the appropriate term in the context they are using it? Would an alternative expression be a bit less misleading?

6. The authors note that the GTEx database "offers less spatial precision compared to the Allen dataset, the data is derived from a larger dataset of donors". Can the authors clarify "less spatial precision"? The difference in number of donors is a whopping difference! Perhaps this needs some more discussion in the manuscript or in the SI.

7. On line 168 the authors use the term 'Central OXTR expression patterns'. Please clarify 'central'.

8. in Figure 2 showing the co-expression Spearman correlations are their p values accompanying these correlations? How should the reader interpret r without significance levels?

9. line 179 "The rank order of central OXT expression was related to the GTEx database rank order, but this fell short of statistical significance ($r_s = 0.42$, $p = 0.23$)". It not only fell short it is simply NOT significant. If there is no significant correlation between the two databases how should the reader understand this. This is an important point and needs clarification.

10. line 186 - specify which 'cortical regions'

11. line 211 "To explore whether these co-expression patterns are not driven by global expression differences...". Clarify this sentence - it's not clear to this referee what is the meant by 'driven by global expression differences'.

12. What method / reference needed was used for the 'exploratory correlation analysis'? mentioned on line 219.

13. These gene abbreviations should be spelled out NTSR2 , GLUD2 GLUD1 when used for first time on line 223-4

14. Line 230 - the authors use 'mental states' and 'cognitive functions'. Are these terms signifying the same thing? It's a bit vague the use of these terms. Can the authors better define these two terms - similarities/dissimilarities.

15. Again in Figure 6 "Cognitive correlates of central oxytocin pathway gene expression" could the authors explain the level of significance of these correlations. Generally Spearman r is accompanied by a p value? What is the meaning of strong relationships without significance levels?

16. line 262 "the OXTR map was ranked among the top 100 out of 20737 genes for several functional imaging maps (i.e., the top 0.5%), demonstrating high specificity". What does it mean to be among the top 100? Needs strict clarification. Additionally, demonstrate what 'high' specificity means? Specificity compared to what?

17. The authors state " PAK7 and RTN4R (Nogo66), which have been associated with schizophrenia 56". The associations with schizophrenia are single? reports and not that convincing?

18. line 312 - Again, the authors use top 100 and it's very unclear how this is to be interpreted. It

sounds like US News and World Report of the top 100 Universities. Not very clear.

19. line 362 " it remains unclear whether the increased expression of OXTR as measured by the presence of its transcript in our human sample is directly relevant for the activation of the receptor during the relevant mental state". This statement needs clarifying since it seems to undermine the main significance of the study?

20. " tentative oxytocin treatment target map that may help facilitate efforts to better understand oxytocin treatment efficacy " Please explain how this practically works? Not clear to me how the map really aids treatment ?

21. The small sample size and the ethnic diversity in the sample is problematic. Can the authors control for ethnicity in the analysis?

Additional Statistical Comments:

1. Isn't the sample size (n = 6 donors from Allen Human Brain Atlas) too small for the study?

2. The statistical analysis of central gene expression data used One-sample t-tests (two-tailed) to assess the 54 left hemisphere regions from six donors samples. Why was chi-square test not used instead?

A chi-square test requires categorical variables, usually only two, but each may have any number of levels. For example, the variables could be ethnic group - White, Black, Asian, American Indian/Alaskan native, Native Hawaiian/Pacific Islander, while a t-test requires two variables; one must be categorical and have exactly two levels, and the other must be quantitative and be estimable by a mean. For example, the two groups could be Republicans and Democrats, and the quantitative variable could be age [Ref : <https://sciencing.com/difference-between-ttest-chi-square-8225095.html>]

3. Why were tools such as WGCNA

(<https://labs.genetics.ucla.edu/horvath/CoexpressionNetwork/Rpackages/WGCNA/>) [Langfelder P and Horvath S, 2008], DiffCoEx [Tesson BM et al, 2010], DICER [Amar D et al, 2013], CoXpress (<http://coxpress.sourceforge.net/>) [Watson M, 2006], DINGO [Ha M et al, 2015], GSCNA [Rahmatallah Y et al, 2014], GSVD [Alter O et al, 2003] not used for co-expression analysis in the study?

4. Why was tools such as PANA [Ponzoni I et al, 2014] not used to study the functional interconnections among the molecular elements of a biological system?

Reviewer #5 (Remarks to the Author):

1) Genetic data was collected only from 6 donor brains. This data is also very heterogeneous as noted by the authors – 3 caucasian males, 1 hispanic female, 2 african-american males. Previous studies have extensively highlighted gender differences in neuroanatomical features which could extend to mRNA expression. Additionally, race and ethnicity could contribute to this making the results inconsistent.

2) Page 7, Line 126 – "Each brain was sampled in 363-946 distinct locations, either in left hemisphere only (n =6), or over both hemispheres (n = 2)...." It seems like the genetic sampling was not consistent across all the donor brains. Also, there were 6 donor samples in all which means that you have left hemisphere only in n=4 and both left and right in n = 2.

3) Given that left and right hemisphere sampling is available only for n = 2, it is unreasonable to say that the gene expression for the right hemisphere samples was highly correlated with the corresponding left hemisphere samples. A more appropriate reason for not using the right

hemisphere samples given the already small sample size would be that not all participants had both right and left hemisphere gene expression data.

4) The highlighted questions in the introduction on Page 5 state that the authors are a) characterizing the anatomical distribution of mRNA expression of OXT, OXTR and CD38 using the Allen Human Brain Atlas, b) Explore putative gene interactions by identifying mRNA maps with overlapping anatomical distributions with target genes (which is presumably the 17) and c) decode mental state relevance of the selected oxytocin genes using reverse inference via fMRI meta-analysis and assess specificity across the 20737 mRNA maps.

a. It is not clear how the authors arrived at the selected gene sample of 17 which they would study from the 20, 737 for aim b). The authors highlight some a priori hypothesis regarding the oxytocin pathways but then not all the genes involved in oxytocin pathways have been included. And then there are some genes from other pathways such as the selected dopamine, muscarinic acetylcholine, and vasopressin gene sets.

b. Also, the reverse inference process seems counter-intuitive to identifying specificity of the genes. For mental state relevance, it would be better if you started with the fMRI meta-analysis areas relevant to social behavior (identified in the abstract) and then look at the mRNA expressions in those regions specific relevant to social behaviors.

c. It seems rather surprising that with neurosynth only social behavior came up as a significant brain state relevant to the regions identified based on the expression maps. Neurosynth is not necessarily selective.

d. Another interesting thing to do would be to query neurosynth to identify regions previously identified to be related to social behaviors and an additional set of regions not identified with social behaviors and look at the oxytocin gene expression in those specific regions to explore the relationship between oxytocin gene expressions and social behavioral states in a more standardized case-control form.

5) The methodology to identify the gene-expression maps seems interesting. However, the individual maps for each of the 6 participants was created in native space. Which means there is significant variability. Another very important factor to note is that in practice, the brain sizes and morphological features encompass a significant amount of variability including brain size and such. The manuscript lacks mention of this completely. Understandably, the inter-subject variability in morphological features are less of a factor for such a small sample size and when considering post-mortem sampling. However, my concern is that it is entirely possible that participants had different brain sizes and thereby introducing variability in the sampling. Additionally, as the authors state on Page 20 Line 385, "If more than one probe for each region was available, we selected the probe with the highest signal-to-noise ratio...". This could also be erroneous since the ignored probes could simply have a suppressed gene expression (signal to noise ratio calculated as mean/standard deviation) in essence just representing the variability across the brain within a subject. This is an important factor that needs to be addressed.

6) The methodological extension from voxel-by-voxel gene expression maps to the 17x17 correlation matrix that the authors mention in the statistical analysis section is unclear. Page 21 Lines 408-417 connecting Page 22 Lines 427 - 439. One can assume that the authors generated a voxel-by-voxel map for each of the 17 genes and then ran a correlation on the averaged (across subjects) maps to get between gene correlations. However, given the standard of the journal, methodological details should not be left to interpretation.

7) The out-of-sample validation was conducted only for 10 distinct brain regions (Page 20, Line 395), whereas for the six participants' brain data 54 regions in the left hemisphere were tested for mRNA expression. The current approach does not seem to represent a systematic selection of brain regions. Seems like the steps for different parts of the study are not common and thereby not necessarily comparable or extendable.

Overall comment to reviewers

We would first like to thank the reviewers for their feedback and queries, which we have addressed below. Please note that while the overarching conclusions of the study remain the same after our revisions, there are some minor differences in the results. These differences occur for two reasons: i) as per Reviewer 3's suggestion we now use an improved method (i.e., differential stability) to select mRNA probes for each gene, ii) since the submission of our manuscript, the NeuroSynth database has been updated with an additional 2965 studies (release 0.7). Thus, to provide an up-to-date and more comprehensive analysis, we now use the updated NeuroSynth database.

Reviewer #1

1. Could the authors provide a rationale for why regions like the striatum are decomposed into sub-units (NAcc/caudate/putamen), but aggregate measures are given for regions like the thalamus and amygdala? It would be very useful to show this data given the heterogeneity of expression profiles across sub-structures.

Response: While there are several options available for brain atlases, we decided to use the AAL atlas, as it provides a sensible compromise between specificity and succinctness. However, we understand that sub-structure expression profiles are also of interest. Thus, we now provide additional expression profiles for eight structures in the hypothalamic region, given that oxytocin is thought to be highly expressed here, and eight structures in the nearby thalamus for comparison. We demonstrate increased expression of OXTR, CD38, and CD38 in the hypothalamus compared to the thalamus, along with increased expression in specific hypothalamic structures, which we report in Figure 2 and in text (Page 11, Line 204):

“...expression patterns within hypothalamic substructures were also summarized using anatomic labels from the Allen dataset (Fig. 2; Supplementary Data 1). Oxytocin pathway gene expression in nearby thalamic substructures were also summarized for comparison (see Supplementary Fig. 7 for anatomical map of substructures). There was significantly higher expression in the hypothalamic structures compared to the thalamic structures for OXTR ($p < .001$), CD38 ($p = .002$) and OXT ($p < .001$). Notably, there was significantly higher expression of OXT mRNA in the PVN, SON, preoptic region, and dorsomedial hypothalamic nucleus substructures compared to mean expression across the hypothalamus and thalamus (FDR corrected $p < .001$).”

We did not include a summary from amygdala substructures as there is fewer available substructures with rRNA expression data compared to the thalamic and hypothalamic substructures.

2. Why is Figure 2B the same as Figure 3A? This redundancy should be removed.

Response: We have now removed this redundancy. This is now only presented as Figure 3b.

3. “To our knowledge, research has yet to apply this approach to identify mental state correlates using brain maps of gene expression.” Please refer to the bioarchive preprint by Fox et al. 2014 (Bridging psychology and genetics using large-scale spatial analysis of

neuroimaging and neurogenetic data). Although not yet published, the work is a significant contribution and is parallel to the analyses in the current manuscript.

Response: Thank you for bringing this preprint to our attention, we have restricted this section of the paper and cite Fox et. al's preprint (Page 17, Line 328):

“To identify cognitive state correlates using brain maps of gene expression, ⁴⁷ we performed quantitative reverse inference via large-scale meta-analysis of functional neuroimaging data using mRNA brain expression maps on voxel-by-voxel left hemisphere brain maps, representing the average of 6 individuals”

4. Could the authors specify which median GTEx values were used. Are these FPKM? RPKM?

Response: We use a log10 value of transcripts per million (TPM) values. We now note this in the methods (Page 26, Line 528):

“Oxytocin pathway median expression values (Log10 transcripts per million) for 10 distinct brain regions (amygdala, anterior cingulate cortex, caudate, cerebellar hemisphere, cerebellum, cortex, frontal cortex, hippocampus, nucleus accumbens, putamen, substantia nigra) were extracted from the Genotype-Tissue Expression (GTEx) project database ³⁹ for independent sample validation.”

5. On page 3 in the introduction, the authors suggest that “While central oxytocin receptor (OXTR) mRNA localization in the rodent brain is well-described, its anatomical distribution across the human brain is poorly understood, as investigations have tended to sample very few brain regions.” I would refer them to the supplementary materials provided in the recently published NatComms paper by Anderson et al., 2018, which reports differential expression across large-scale brain networks, or the earlier work they cite from Steve Chang's lab (Dal Monte et al., 2017), which presents expression across a dense sampling of anatomical regions.

Response: Thank you for bringing the paper from Anderson et al to our attention, which was published after we submitted our manuscript. In our revision we now mention that Anderson et al (reference #19 in our paper) report differential expression in the limbic system for OXTR. We also mention results from the Dal Monte et al paper (reference #19 in our paper; Page 3, Line 70):

“While more recent work has exclusively examined whole-brain distribution of OXTR genes against a limited set of genes ¹⁸ and overexpression in broad functional brain networks (e.g., limbic network) ¹⁹, researchers have yet to explore OXTR's associations with all protein-coding genes or with gene expression patterns associated with specific cognitive states.”

Reviewer #2

1. Abstract - the lines 25-29 that suggests why this study is potentially clinically relevant seems quite artificial and superficial. The article is of interest regardless of its supposed clinical implications. I urge the authors to delete these superfluous lines.

Response: We have deleted these lines.

2. The concluding sentence in the Abstract - "These results provide a proof-of-principle demonstration of corresponding mental states with gene expression" - is quite weak and not very informative regarding the take home message for this manuscript. I strongly suggest revising the concluding statement.

Response: We have revised the concluding sentence of the abstract to be more informative regarding the central message of the paper (Page 2, Line 43):

"Altogether, these analyses indicate that the oxytocin signaling system interacts with dopaminergic and muscarinic acetylcholine signaling to modulate cognitive state processes involved in complex human behaviors."

3. I suggest deleting lines 62-64 (in line with my Comment 1).

Response: We have deleted these lines.

4. The authors mention the opioid pathway "evidence of co-expression with 86 OXTR mRNA and the μ and κ types opioid receptor mRNA" and nevertheless don't examine its interactions with the oxytocin system. Why?

Response: We now examine the co-expression of opioid receptor genes with the oxytocin pathway system, which we describe in the updated paper (Page 7, Line 138):

"Four other selected sets of mRNA, which are thought to co-express with oxytocin pathway mRNA and modulate social behavior, were also of specific interest: A dopaminergic set (DRD1, DRD2, DRD3, DRD4, DRD5, COMT, DAT1)^{22, 23, 37}, a muscarinic acetylcholine set (CRHM1, CRHM2, CRHM3, CRHM4, CRHM5)^{16, 24}, a vasopressin set (AVPR1A, AVPR1B)^{33, 38} and an opioid set (OPRM1, OPRD1, OPRK1)¹⁸"

We found that the co-expression of the oxytocin pathway and opioid receptor pathway do not cluster together in the same network module (Fig. 3a).

5. The authors use the term "out-of-sample" validation. This term seems to have very specific definition regarding statistical model testing http://ec.europa.eu/eurostat/statistics-explained/index.php/Glossary:In-sample_vs._out-of-sample_forecasts

I wonder if this is the appropriate term in the context they are using it? Would an alternative expression be a bit less misleading?

Response: We now use an alternative expression for this analysis throughout the manuscript, "Independent sample validation"

6. The authors note that the GTEx database "offers less spatial precision compared to the Allen dataset, the data is derived from a larger dataset of donors". Can the authors clarify "less spatial precision"? The difference in number of donors is a whopping difference! Perhaps this needs some more discussion in the manuscript or in the SI.

Response: Our original intention was to state that there was less spatial resolution in the GTEx dataset, as this dataset only provides mRNA expression data from 10 brain regions (Compared to 363+ sampling regions in the Allen dataset). We now use the term “spatial resolution” in the manuscript, as follows (Page 26, Line 533):

“Although this dataset offers less spatial resolution compared to the Allen dataset, the data is derived from a larger dataset of donors (mean sample size for mRNA expression across brain regions = 131.7, range = 88 to 173).”

There is a substantial difference in the number of donors between databases, however we now demonstrate using differential stability that genes expression patterns from the Allen dataset are highly reproducible from donor-to-donor (See query 21 below). We mention the difference in the number of donors as a study limitation (Page 24, Line 476):

“We also independently validated general gene expression patterning in the larger GTEx dataset, however, it is worth noting that this comparison is somewhat limited due to fewer brain tissue sampling sites in the GTEx dataset compared to the Allen dataset”.

7. On line 168 the authors use the term 'Central OXTR expression patterns'. Please clarify 'central'.

Response: We meant to convey OXTR expression patterns within the brain. We have now reworded this section (Page 9, Line 176):

“OXTR expression patterns in the brain were comparable to....”

We have also reworded other instance of the term “central” throughout the manuscript when referring the brain in general.

8. In Figure 2 showing the co-expression Spearman correlations are their p values accompanying these correlations? How should the reader interpret r without significance levels?

Response: We now present significance levels in this matrix (Fig. 3a). Of note, all correlations within the OXTR/CD38 cluster were statistically significant.

9. Line 179 "The rank order of central OXT expression was related to the GTEx database rank order, but this fell short of statistical significance ($r_s = 0.42$, $p = 0.23$)". It not only fell short it is simply NOT significant. If there is no significant correlation between the two databases how should the reader understand this. This is an important point and needs clarification.

Response: We now make it clearer that this relationship was not statistically significant (Page 10, Line 195):

“OXT expression patterns in the Allen database were not significantly associated to the GTEx expression patterns ($r_s = 0.22$, $p = 0.54$).”

We clarify these results and temper conclusions regarding the expression of OXT in the discussion section (Page 21, Line 398):

“Expression patterns of OXT in the Allen database were less stable from donor-to-donor and were not significantly related to the expression patterns in the GTEx database, which limits the generalizability of the results relating to this specific gene.”

10. Line 186 - specify which 'cortical regions'

Response: We now make this sentence clearer (Page 11, Line 199):

“Of note, DRD1, DRD2, CHRM4, and OPRK1 genes were highly expressed in subcortical regions, including the caudate, pallidum, putamen, and thalamus (Supplementary Data 1)”

11. Line 211 "To explore whether these co-expression patterns are not driven by global expression differences...". Clarify this sentence - it's not clear to this referee what is the meant by 'driven by global expression differences'.

Response: We have now clarified this sentence (Page 13, Line 248):

“To explore whether these co-expression patterns are not driven by gene expression differences between cortical and non-cortical areas, we visualized the similarities in expression patterns based on cortical areas only for comparison against whole brain patterns (Fig. 3c).

12. What method / reference needed was used for the 'exploratory correlation analysis'? mentioned on line 219.

Response: To identify single genes associated with OXTR, CD38, and OXT, we used Spearman's correlations to identify which gene expression maps most closely approximated these maps. To improve clarity, we have removed the term “exploratory” (Page 15, Line 283):

“Correlation analyses using Spearman's coefficient with all available 20737 protein coding gene probes in the Allen Human Brain Atlas and each of the three oxytocin pathway gene probes are summarized in Table 1, with the top 10 strongest positive and top 10 strongest negative correlations (the full set of correlations with the remaining 20736 genes are presented in Supplementary Data 2).

13. These gene abbreviations should be spelled out NTSR2 , GLUD2 GLUD1 when used for first time on line 223-4

Response: We now spell out these abbreviations (Page 15, Line 288);

“Notably, among the top ten most positive mRNA map correlations for OXTR were Neurotensin Receptor 2 (NTSR2; $r_s = 0.78$), Glutamate dehydrogenase 2 (GLUD2; $r_s = 0.78$), and Glutamate dehydrogenase 1 (GLUD1; $r_s = 0.77$),

14. Line 230 - the authors use 'mental states' and 'cognitive functions'. Are these terms signifying the same thing? It's a bit vague the use of these terms. Can the authors better define these two terms - similarities/dissimilarities.

Response: We now use the term “cognitive states” throughout the manuscript, instead of “mental states” or “cognitive functions”. We also define the term “cognitive states” at the first opportunity in the manuscript, as follows (Page 17, Line 314):

“The NeuroSynth framework ³⁴ has extracted text and neuroimaging data from 14371 fMRI studies (release 0.7). While this tool can be used to create meta-analytic brain activation maps for specific cognitive states (e.g., stress) using forward inference, it can also be used to “decode” cognitive states, given an activation map, via reverse inference. Here, the term “cognitive state” refers to the neural processing that occurs response to a specific stimulus (e.g., pain) or that underlies a psychological process (e.g., learning).

15. Again in Figure 6 "Cognitive correlates of central oxytocin pathway gene expression" could the authors explain the level of significance of these correlations. Generally Spearman r is accompanied by a p value? What is the meaning of strong relationships without significance levels?

Response: We present now these p-values in Table 2 and we make these results more prominent in the manuscript (Page 19, Line 362):

“Notably, the OXTR map’s relationship with the cognitive state maps of “sexual”, “motivation”, “incentive”, and “anxiety” were all ranked among the top 0.5% strongest associations out of 20737 genes and were statistically significant ($p < .001$) Fig. 8B; Table 2), The “taste”, “stress”, “reward”, “monetary”, “fear”, and “emotional” cognitive state maps were within the top 2.5% of all associations with OXTR expression and also statistically significant ($p < .001$; Fig. 8B; Table 2).

16. Line 262 "the OXTR map was ranked among the top 100 out of 20737 genes for several functional imaging maps (i.e., the top0.5%), demonstrating high specificity". What does it mean to be among the top 100? Needs strict clarification. Additionally, demonstrate what 'high' specificity means? Specificity compared to what?

Response: For this analysis, we assessed the relationship between a set of given cognitive state functional activity maps and the expression maps of 20737 genes. This yielded a set of correlation coefficients that we ordered from the strongest positive correlation to the strongest negative correlation. The top 0.5% genes that we referred to were the gene expression maps with the top 0.5% highest positive correlations with a respective cognitive state map, which approximately corresponds to the top 100 out of 20737 associations. The top 0.5% can be considered an arbitrary threshold, but we selected it simply as a frame of reference. Readers can inspect the histograms in Figure 8 and the data in Table 2 for reference.

In regards to specificity: We first assessed which cognitive states were the most highly correlated with the genes of interest. Despite statistical significance, there could have been many other genes that were also more highly expressed in these regions. For instance, if we found a statistically significant correlation between the OXTR gene expression map and the “Reward” cognitive state map, perhaps there were thousands of other gene expression maps that had a stronger correlation with the “Reward” cognitive state map. By calculating the correlation between all 20737 protein coding genes and these resting state maps, it’s possible to assess whether OXTR has a uniquely high level of expression in these regions, or whether there’s a substantial portion of other genes whose expression patterns have a stronger

association with a given cognitive state. We now clarify what being among the top 100 genes means in the manuscript (Page 19, Line 362):

“Notably, the OXTR map’s relationship with the cognitive state maps of “sexual”, “motivation”, “incentive”, and “anxiety” were all ranked among the top 0.5% strongest associations out of 20737 genes and were statistically significant ($p < .001$) Fig. 8B; Table 2), The “taste”, “stress”, “reward”, “monetary”, “fear”, and “emotional” cognitive state maps were within the top 2.5% of all associations with OXTR expression and also statistically significant ($p < .001$; Fig. 8B; Table 2). In other words, not only were these cognitive state maps the most highly correlated with the OXTR expression map, but these correlations with specific cognitive states were amongst the strongest correlations across all 20373 protein coding genes”.

17. The authors state " PAK7 and RTN4R (Nogo66), which have been associated with schizophrenia 56". The associations with schizophrenia are single? reports and not that convincing?

Response: We now specify that these were only single reports that require replication in independent samples (Page 22, Line 426):

“While these studies suggest intriguing links between specific genes and psychiatric illnesses, it is important to note that these are largely single report associations that require replication in independent samples”

18. Line 312 - Again, the authors use top 100 and it's very unclear how this is to be interpreted. It sounds like US News and World Report of the top 100 Universities. Not very clear.

Response: As we outlined in response to query #16, this ranking refers to the correlation between a gene expression map and a given cognitive state map. We have reworded these sentences to improve interpretation (Page 19, Line 356):

“Notably, the OXTR map’s relationship with the cognitive state maps of “sexual”, “motivation”, “incentive”, and “anxiety” were all ranked among the top 0.5% strongest associations out of 20737 genes and were statistically significant ($p < .001$) Fig. 8B; Table 2), The “taste”, “stress”, “reward”, “monetary”, “fear”, and “emotional” cognitive state maps were within the top 2.5% of all associations with OXTR expression and also statistically significant ($p < .001$; Fig. 8B; Table 2). In other words, not only were these cognitive state maps the most highly correlated with the OXTR expression map, but these correlations with specific cognitive states were amongst the strongest correlations across all 20373 protein coding genes”.

19. Line 362 “It remains unclear whether the increased expression of OXTR as measured by the presence of its transcript in our human sample is directly relevant for the activation of the receptor during the relevant mental state". This statement needs clarifying since it seems to undermine the main significance of the study?

Response: Animal research has shown that increased oxytocin receptor gene expression levels directly modulate social behavior and that increased oxytocin receptor gene expression is

associated with increased oxytocin binding. We have clarified this statement to better reflect that this has yet to be shown in humans due to practical reasons (Page 25, Line 492):

“...despite animal models demonstrating that OXTR expression levels directly modulate social behavior^{13,14} and that increased OXTR gene expression is associated with increased oxytocin binding, research is yet to demonstrate whether the increased expression of OXTR (as measured by the presence of its transcript) in humans is associated with the activation of the receptor during specific cognitive states.”

20. "tentative oxytocin treatment target map that may help facilitate efforts to better understand oxytocin treatment efficacy " Please explain how this practically works? Not clear to me how the map really aids treatment?

Response: One of the reasons we conducted this study was to address an important gap in the literature raised by Thomas Insel in a perspective article from the US National Institute of Mental Health (Biol. Psychiatry, 79(3), 2016). In this article, Insel states (pg. 154), “To improve the rigor and reproducibility of [oxytocin] clinical trials, each study must demonstrate target engagement, which means evidence of activation of a proposed mechanism at a clinically effective dose. Occupancy of oxytocin receptors in a specific brain area would be an excellent example of target engagement, but so far there is no positron emission tomography ligand and no compelling evidence of oxytocin receptors in the human brain.”

Here, we have not only demonstrated evidence of oxytocin receptors in the human brain but also the location of these receptors along with their cognitive state correlates. We have reworded this section to better reflect our intentions (Page 25, Line 498):

“To translate oxytocin to the clinic, research needs to demonstrate engagement of drug targets, such as OXTR occupancy reflected by regional brain activity changes⁸³. Without precise targets, it is unclear whether non-significant effects of intranasal oxytocin, beyond insufficient statistical power⁸⁴, are due to an inefficacious drug or misidentified drug targets. By identifying precise oxytocin pathway targets in the human brain and the cognitive state correlates of these oxytocin pathway gene distribution, our study may help facilitate efforts to better understand oxytocin treatment efficacy as it identifies targets for oxytocin receptor engagement, which can facilitate dose-ranging studies^{82,83}.”

21. The small sample size and the ethnic diversity in the sample is problematic. Can the authors control for ethnicity in the analysis?

Response: Given sample size limitation, we cannot effectively control for ethnicity within our analyses. However, we took two approaches to better understand the potential role of the small sample size and ethnicity heterogeneity. First, we assessed the reproducibility of gene expression patterning across the six brains in the sample by using the concept of differential stability, which has previously been applied to the same dataset (Hawrylycz et al., 2015, *Nat. Neurosci.*, 18). High differential stability would suggest that gene expression patterning is reproducible from donor-to-donor. We found that both *OXTR* and *CD38* had differential stability values in the top decile of all 20737 protein coding genes in the dataset. Genes with high differential stability were also shown by Hawrylycz and colleagues (2015) to be highly biologically relevant. Indeed, in our updated analysis we confirmed that both *OXTR* and *CD38* belong to an oxytocin pathway gene module which is significantly enriched in GWAS of waist-hip ratio, as well as several behavioral and cognitive state processes (e.g., behavioral response

to stimulus, cognition, feeding behavior). We report these results as follows (Page 16, Line 303):

We found that both OXTR and CD38 had differential stability values in the top decile of all 20737 protein coding genes in the dataset (Fig. 6a), indicating that gene expression patterning is reproducible, regardless of individual differences, such as ethnicity and sex.

Second, we also calculated donor-to-donor associations of gene expression patterns, revealing statistically significant associations. We report these results as follows (Page 16, Line 306):

“We also calculated donor-to-donor associations of gene expression patterns, revealing statistically significant associations (FDR corrected $p < .0001$) for OXTR, CD38, and OXT expression patterns between all donors (Figs. 6b-d).”

Thus, despite the small sample size and ethnicity heterogeneity, gene expression of OXTR and CD38 was highly reproducible.

22. Isn't the sample size (n = 6 donors from Allen Human Brain Atlas) too small for the study?

Response: As described above in query #21, we show a high degree of donor-to-donor spatial precision in microarray measures, suggesting that the sample size was sufficient for the study. Moreover, the Allen Human Brain Atlas is the best dataset available to explore the relationship between gene expression patterns and cognitive state neural patterns due to its high spatial resolution.

23. The statistical analysis of central gene expression data used One-sample t-tests (two-tailed) to assess the 54 left hemisphere regions from six donor samples. Why was chi-square test not used instead?

A chi-square test requires categorical variables, usually only two, but each may have any number of levels. For example, the variables could be ethnic group - White, Black, Asian, American Indian/Alaskan native, Native Hawaiian/Pacific Islander, while a t-test requires two variables; one must be categorical and have exactly two levels, and the other must be quantitative and be estimable by a mean. For example, the two groups could be Republicans and Democrats, and the quantitative variable could be age [Ref : <https://sciencing.com/difference-between-ttest-chi-square-8225095.html>]

Response: We used a one-sample t-test as we are interested in whether gene expression was significantly different in a given brain region compared to mean expression across all regions. This comparison against mean global expression is a common approach when assessing gene expression data from the Allen Human Brain Atlas (e.g. Dal Monte et al., *PNAS*, 2017, DOI: 10.1073/pnas.1702725114).

24. Why were tools such as WGCNA (<https://labs.genetics.ucla.edu/horvath/CoexpressionNetwork/Rpackages/WGCNA/>) [Langfelder P and Horvath S, 2008], DiffCoEx [Tesson BM et al, 2010], DICER [Amar D et al, 2013], CoXpress (<http://coxpress.sourceforge.net/>) [Watson M, 2006], DINGO [Ha M et al, 2015], GSCNA [Rahmatallah Y et al, 2014], GSVD [Alter O et al, 2003] not used for co-expression analysis in the study?

Response: We now use WGCNA to construct co-expression of the full oxytocin pathway geneset in the brain. We outline our procedure in the methods (Page 30, Line 605). We identified co-expression module containing *OXTR* and *CD38* (22 genes in total), which we describe in the (Page 13, Line 256):

*“To uncover co-expression patterns of the full oxytocin signaling pathway (94 genes), we applied weighted gene co-expression network analysis using the WGCNA R package⁴². This analysis identified a co-expression module containing both *OXTR* and *CD38*, among 28 genes in total (Fig. 4a, Supplementary Data 1; Supplementary Fig. 9). Submission of the *OXTR/CD38* module to FUMA revealed that the this geneset module is enriched in GWAS catalog reported genes⁴³ for waist-hip ratio adjusted for BMI and smoking ($p = 1.13 \times 10^{-2}$). This *OXTR/CD38* module was also enriched in genesets reported in the Molecular Signatures Database⁴⁴ that are associated with several behavioral and cognitive state processes including feeding behavior ($p = 5.83 \times 10^{-3}$), cognition ($p = 7.44 \times 10^{-3}$), and behavioral response to stimuli ($p = 1.05 \times 10^{-2}$). This *OXTR/CD38* module was also enriched in brain and breast tissue (Bonferroni corrected p -value $< .05$; Fig. 4b). Altogether, these results suggest that the identified *OXTR/CD38* module is biological meaningful.”*

25. Why was tools such as PANA [Ponzoni I et al, 2014] not used to study the functional interconnections among the molecular elements of a biological system?

Response: While PANA is an interesting approach, we did not examine the functional interconnections among the molecular elements of a biological system as this is beyond the scope of the article, which primarily focuses on the spatial gene expression patterns in the brain and how these patterns related to cognitive states. Alternatively, we now use FUMA (Watanabe et al., *Nat. Comms.*, 8: 1826) to assess the functional significance of the oxytocin pathway network identified using WGCNA. As mentioned above in response #25, we identified statistically significant (FDR corrected) enrichment for waist-hip ratio GWAS along gene sets associated with several cognitive states (e.g., behavioural response to stimulus, cognition, feeding behavior).

Reviewer #5

1. Genetic data was collected only from 6 donor brains. This data is also very heterogeneous as noted by the authors – 3 caucasian males, 1 hispanic female, 2 african-american males. Previous studies have extensively highlighted gender differences in neuroanatomical features which could extend to mRNA expression. Additionally, race and ethnicity could contribute to this making the results inconsistent.

Response: We now provide additional analysis demonstrating that expression patterns for *OXTR* and *CD38* were highly stable between participants. We addressed the same query to another reviewer, however for convenience we reproduce our response here:

We took two approaches to better understand the potential role of the small sample size and ethnicity heterogeneity. First, we assessed the reproducibility of gene expression patterning across the six brains in the sample by using the concept of differential stability, which has previously been applied to the same dataset (Hawrylycz et al., 2015, *Nat. Neurosci.*, 18). High

differential stability would suggest that gene expression patterning is reproducible, regardless of ethnicity. We found that both OXTR and CD38 had differential stability values in the top decile of all 20737 protein coding genes in the dataset. Genes with high differential stability were also shown by Hawrylycz and colleagues (2015) to be highly biologically relevant. Indeed, in our updated analysis we confirmed that both OXTR and CD38 belong to oxytocin pathway gene module which is significantly enriched in GWAS of waist-hip ratio, as well as several behavioural and cognitive state processes (e.g., general behavior, cognition, feeding behavior). We report these results as follows (Page 16, Line 303):

“We found that both OXTR and CD38 had differential stability values in the top decile of all 20737 protein coding genes in the dataset (Fig. 6a), indicating that gene expression patterning is reproducible, regardless of individual differences, such as ethnicity and sex.”

Second, we also calculated donor-to-donor associations of gene expression patterns, revealing statistically significant associations. We report these results as follows (Page 16, Line 306):

“We also calculated donor-to-donor associations of gene expression patterns, revealing statistically significant associations (FDR corrected $p < .0001$) for OXTR, CD38, and OXT expression patterns between all donors (Figs. 6b-d). .”

Thus, despite the small sample size and ethnicity heterogeneity, gene expression of OXTR and CD38 was highly reproducible.

2. Page 7, Line 126 – “Each brain was sampled in 363-946 distinct locations, either in left hemisphere only (n =6), or over both hemispheres (n = 2)...” It seems like the genetic sampling was not consistent across all the donor brains. Also, there were 6 donor samples in all which means that you have left hemisphere only in n=4 and both left and right in n = 2.

Response: Despite inconsistent genetic sampling across all donor brains, we demonstrated high differential stability in our voxel-to-voxel maps (as described above in response #1), which were created to interpolate “missing” voxels, indicating reproducible gene expression patterning from donor-to-donor. In our analysis we only use left hemisphere samples (n = 6).

3. Given that left and right hemisphere sampling is available only for n = 2, it is unreasonable to say that the gene expression for the right hemisphere samples was highly correlated with the corresponding left hemisphere samples. A more appropriate reason for not using the right hemisphere samples given the already small sample size would be that not all participants had both right and left hemisphere gene expression data.

Response: We have now reworded this section, removing the correlational analysis (Page 7, Line 131):

“Analyses were performed on left hemisphere samples due to a larger sample size.”

4. The highlighted questions in the introduction on Page 5 state that the authors are a) characterizing the anatomical distribution of mRNA expression of OXT, OXTR and CD38 using the Allen Human Brain Atlas, b) Explore putative gene interactions by identifying mRNA maps with overlapping anatomical distributions with target genes

(which is presumably the 17) and c) decode mental state relevance of the selected oxytocin genes using reverse inference via fMRI meta-analysis and assess specificity across the 20737 mRNA maps.

4a. It is not clear how the authors arrived at the selected gene sample of 17 which they would study from the 20, 737 for aim b). The authors highlight some a priori hypothesis regarding the oxytocin pathways but then not all the genes involved in oxytocin pathways have been included. And then there are some genes from other pathways such as the selected dopamine, muscarinic acetylcholine, and vasopressin gene sets.

Response: We now provide a more in-depth justification for why we selected these specific genes. Our manuscript now combines both hypothesis-driven and exploratory research questions. For our hypothesis driven analysis, we chose the 17 genes (which is now 20 genes upon the suggestion of another reviewer) as they have been previously proposed to interact with the oxytocin system in the context of human behavior (Page 7, Line 136):

“Of primary interest were the three oxytocin pathway genes that have been associated with social behavior in both animal and human research: OXTR, CD38, and OXT⁶. Four other selected sets of mRNA, which are thought to co-express with oxytocin pathway mRNA and modulate social behavior, were also of specific interest: A dopaminergic set (DRD1, DRD2, DRD3, DRD4, DRD5, COMT, DAT1)^{22, 23, 37}, a muscarinic acetylcholine set (CRHM1, CRHM2, CRHM3, CRHM4, CRHM5)^{16, 24}, a vasopressin set (AVPR1A, AVPR1B)^{33, 38} and an opioid set (OPRM1, OPRD1, OPRK1)¹⁸.

We compare expression of OXTR, CD38, and OXT against all 20737 genes and report the top 10 most positive and top 10 most negative correlations (Page 15, Line 283):

“Correlation analyses using Spearman’s coefficient with all available 20737 protein coding gene probes in the Allen Human Brain Atlas and each of the three oxytocin pathway gene probes are summarized in Table 1, with the top 10 strongest positive and top 10 strongest negative correlations (the full set of correlations with the remaining 20736 genes are presented in Supplementary Data 2). The p-values for these correlations were all < 0.001. Notably, among the top ten most positive mRNA map correlations for OXTR were Neurotensin Receptor 2 (NTSR2; $r_s = 0.78$), Glutamate dehydrogenase 2 (GLUD2; $r_s = 0.78$), and Glutamate dehydrogenase 1 (GLUD1; $r_s = 0.77$), and the top ten most positive correlations with CD38 included NTSR2 ($r_s = 0.82$), C12orf39 (Spexin; $r_s = 0.81$), Phosphoserine Aminotransferase 1 (PSAT1; $r_s = 0.79$), and GLUD1 ($r_s = 0.78$).”

We also conduct an additional analysis of the full oxytocin pathway geneset (n = 94 genes) using weighted gene co-expression analysis to assign genes to co-expression modules, which revealed that the module containing OXTR and CD38 is enriched in GWAS catalog reported genes for waist-hip ratio and also enriched in genesets associated with feeding behavior, cognition, and behavioral response to stimulus. We describe these results in the manuscript (Page 13, Line 260):

“Submission of the OXTR/CD38 module to FUMA revealed that the this geneset module is enriched in GWAS catalog reported genes⁴³ for waist-hip ratio adjusted for BMI and smoking ($p = 1.13 \times 10^{-2}$). This OXTR/CD38 module was also enriched in genesets reported in the Molecular Signatures Database⁴⁴ that are associated with several behavioral and cognitive state processes including feeding behavior ($p = 5.83 \times 10^{-3}$), cognition ($p = 7.44 \times 10^{-3}$), and

behavioral response to stimuli ($p = 1.05 \times 10^{-2}$). This OXTR/CD38 module was also enriched in brain and breast tissue (Bonferroni corrected p -value $< .05$; Fig. 4b).

4b. Also, the reverse inference process seems counter-intuitive to identifying specificity of the genes. For mental state relevance, it would be better if you started with the fMRI meta-analysis areas relevant to social behavior (identified in the abstract) and then look at the mRNA expressions in those regions specific relevant to social behaviors.

Response: We now include an analysis exploring mRNA in brain regions implicated in social behavior (see response 4d below).

4c. It seems rather surprising that with neurosynth only social behavior came up as a significant brain state relevant to the regions identified based on the expression maps. Neurosynth is not necessarily selective.

Response: The reverse inference approach selects the highest probability that a given map is associated with a large corpus of cognitive states. This helps rule out whether a given activation map is better explained by other cognitive states. In our analysis, we selected the top 5 cognitive state brain activation maps that were most highly correlated with gene expression maps. To determine the relative strength of the oxytocin pathway gene associations, we also visualised the ranking of these correlations against all 20737 protein coding genes. OXTR and CD38 were among the top 0.5% of correlation coefficients for several cognitive state maps. Therefore, not only was OXTR and CD38 significantly associated with the identified cognitive states, these were among the strongest correlations compared to all protein coding genes.

4d. Another interesting thing to do would be to query neurosynth to identify regions previously identified to be related to social behaviors and an additional set of regions not identified with social behaviors and look at the oxytocin gene expression in those specific regions to explore the relationship between oxytocin gene expressions and social behavioral states in a more standardized case-control form.

Response: This is good suggestion. In the revised version of the manuscript we present results from a comparison of gene expression of OXTR, CD38, and OXT in areas related to social behavior with areas not identified with social behaviours. We found increased mRNA expression of OXTR and CD38 in brain regions related to social behavior, which we now report in the results (Page 20, Line 374):

“Additionally, OXTR ($p = .0002$; Fig. 8c) and CD38 ($p = .006$; Fig. 8d), had significantly greater mRNA expression in brain regions associated with social behavior compared to brain regions associated with non-social cognitive states (Supplementary Data 1). There was no significant difference in OXT mRNA expression ($p = .14$; Fig 8e) between these social and non-social brain regions.

5. The methodology to identify the gene-expression maps seems interesting. However, the individual maps for each of the 6 participants was created in native space. Which means there is significant variability. Another very important factor to note is that in practice, the brain sizes and morphological features encompass a significant amount of variability including brain size and such. The manuscript lacks mention of this completely. Understandably, the inter-subject variability in morphological features are less of a factor

for such a small sample size and when considering post-mortem sampling. However, my concern is that it is entirely possible that participants had different brain sizes and thereby introducing variability in the sampling. Additionally, as the authors state on Page 20 Line 385, “If more than one probe for each region was available, we selected the probe with the highest signal-to-noise ratio...”. This could also be erroneous since the ignored probes could simply have a suppressed gene expression (signal to noise ratio calculated as mean/standard deviation) in essence just representing the variability across the brain within a subject. This is an important factor that needs to be addressed.

Response: As correctly noted by the reviewer, we created expression maps in native space, and then used ANTs non-linear normalization to align brains to the MNI152 template. We now take a different approach to probe selection in which we select the probe with the least donor-to-donor variability by using differential stability. As we note to another reviewer, the concept of differential stability has previously been applied to the same dataset (Hawrylycz et al., 2015, Nat. Neurosci., 18). By selecting probes using differential stability, we are selecting the probe with the most reproducible gene expression patterning from donor-to-donor.

We now mention differences in morphological features from donor-to-donor as a study limitation (Page 24, Line 469):

“There are some important limitations worth noting. First, given the nature of human brain tissue collection for mRNA studies, the sample size was small and the donor group was variable in regards to age and ethnicity. These issues, along with individual differences in brain sizes and morphological features, may have introduced variability in sampling. However, there are several reasons to believe that donor-to-donor variability did not influence the outcomes of the study.”

6. The methodological extension from voxel-by-voxel gene expression maps to the 17x17 correlation matrix that the authors mention in the statistical analysis section is unclear. Page 21 Lines 408-417 connecting Page 22 Lines 427 - 439. One can assume that the authors generated a voxel-by-voxel map for each of the 17 genes and then ran a correlation on the averaged (across subjects) maps to get between gene correlations. However, given the standard of the journal, methodological details should not be left to interpretation.

Response: We now provide greater detail regarding our methods as per the following (Page 28, Line 577):

“Based on raw expression data, we generated a 20×20 correlation matrix reflecting the spatial Spearman’s correlation for each donor for each mRNA map pair including the following selected genes (Oxytocin pathway set: OXTR, CD38, OXT; Dopaminergic set: DRD1, DRD2, DRD3, DRD4, DRD5, COMT, DAT1; muscarinic acetylcholine set: CRHM1, CRHM2, CRHM3, CRHM4, CRHM5; vasopressin set: AVPR1A, AVPR1B; opioid set: OPRM1, OPRD1, OPRK1). Next, we computed the average correlation in each cell of the correlation matrix across the 6 donors and clustered the mRNA maps using the complete linkage method⁸⁵ to assess co-expression patterns. To assess putative gene-gene interactions beyond selected gene sets, we computed the spatial Spearman’s correlations between all available protein coding mRNA probes ($n = 20737$) and three oxytocin pathway mRNA probes. As more than one mRNA probe may have been available for a single gene, we selected the probe with the highest differential stability value. Unlike Pearson’s r , which is the correlate statistic presented on the

Allen Human Brain Atlas web interface, Spearman's correlation coefficient is less sensitive to outliers and non-normally distributed data."

7. The out-of-sample validation was conducted only for 10 distinct brain regions (Page 20, Line 395), whereas for the six participants' brain data 54 regions in the left hemisphere were tested for mRNA expression. The current approach does not seem to represent a systematic selection of brain regions. Seems like the steps for different parts of the study are not common and thereby not necessarily comparable or extendable.

Response: We used the Allen human brain dataset for our main analysis as it offered the richest spatial resolution, which also facilitated our analysis of cognitive state pattern associations. To help validate the expression patterns uncovered in the Allen human brain atlas we performed a comparison with the GTEx database. However, only expression values from ten specific brain regions are available from GTEx. Thus, we calculated expression data in the Allen human brain atlas regions available in the GTEx database. We now note in the discussion that there is some discrepancy between the Allen and GTEx databases (Page 24, Line 476):

"We also independently validated general gene expression patterning in the larger GTEx dataset, however, it is worth noting that this comparison is somewhat limited due to fewer brain tissue sampling sites in the GTEx dataset compared to the Allen dataset."

Reviewer #1 (Remarks to the Author):

In the present manuscript the authors characterize the distribution of OXT, OXTR, and CD38 mRNA across the human brain and examine associations between gene expression patterns and mental states through fMRI meta-analysis (NeuroSynth).

I reviewed the initial submission of this work. The authors have been quite responsive to my prior feedback on their manuscript, my remaining concerns/comments which I detail below.

In the last sentence of the abstract, the authors imply a causal link between the oxytocin signaling system and the modulation of cognitive state processes involved in complex human behaviors. The current analyses show spatial overlap between profiles of gene expression and fMRI meta-analytic maps. However, while useful in hypothesis generation, these data are not sufficient to make causal claims.

I might be misinterpreting the present analyses but it seems that the authors are citing a accumbens/OXT connection, but ACB is not a region analyzed in Figure 1.

Minor point, but Figure 2 d-f would benefit from region labels. Relatedly, I am not sure that the current version of figure 2 as that meaningful. As noted in the figure, the hypothalamic expression of OXTR/CD38 are higher in hypothalamus compares to thalamus. Can the authors provide their rationale for choosing the thalamus as the control region? Figure 1 already shows that it has lower than average expression of OXT markers.

In Fig 3, I may be misreading this and I appreciate the relative strength of the correlations in panels B/C, but don't these show that the DA/ACh gene sets do not demonstrate globally strong correlations to oxytocin markers? A genome-wide WGCNA approach may be more appropriate here, but would likely not show that these gene groups are highly related to one another.

Figure 5 seems to be a bit redundant in making the point that OXTR/CD38 are correlated to each other. Perhaps shift it to the supplement?

The authors note: "Notably, among the top ten most positive mRNA map correlations for OXTR were Neurotensin Receptor 2 (NTSR2; $rs = 0.78$), Glutamate dehydrogenase 2 (GLUD2; $rs = 0.78$), and Glutamate dehydrogenase 1 (GLUD1; $rs = 0.77$), and the top ten most positive correlations with CD38 included NTSR2 ($rs = 0.82$), C12orf39 (Spexin; $rs = 0.81$), Phosphoserine Aminotransferase 1 (PSAT1; $rs = 0.79$), and GLUD1($rs = 0.78$)." It may help the readers parse the analyses if they provide some theoretical context detailing what makes these relations "notable."

Can the authors include a brief note acknowledging that OXT transcripts are only expected to be present in OXT producing cells, which may be why it does not show whole-brain correspondence to OXTR/CD38. These data indicate the receptors are transcribed at site of uptake, but OXT is produced in known oxytonergic hypothalamic regions.

Reviewer #4 (Remarks to the Author):

The authors have successfully answered the referees comments. In my opinion the article is now acceptable for publication.

Reviewer #5

[Was not available to re-review, but both remaining reviewers indicated that the authors had addressed Reviewer #5's earlier concerns adequately.]

Response to reviewer

In the present manuscript the authors characterize the distribution of OXT, OXTR, and CD38 mRNA across the human brain and examine associations between gene expression patterns and mental states through fMRI meta-analysis (NeuroSynth).

I reviewed the initial submission of this work. The authors have been quite responsive to my prior feedback on their manuscript, my remaining concerns/comments which I detail below.

1. In the last sentence of the abstract, the authors imply a causal link between the oxytocin signaling system and the modulation of cognitive state processes involved in complex human behaviors. The current analyses show spatial overlap between profiles of gene expression and fMRI meta-analytic maps. However, while useful in hypothesis generation, these data are not sufficient to make causal claims.

Response: We now temper the last sentence of the abstract, which reads as follows:

“The oxytocin signaling system may interact with dopaminergic and muscarinic acetylcholine signaling to modulate cognitive state processes involved in complex human behaviors”

2. I might be misinterpreting the present analyses but it seems that the authors are citing a accumbens/OXT connection, but ACB is not a region analyzed in Figure 1.

Response: We did not specifically evaluate mRNA expression in the Nucleus Accumbens in our primary analysis, which is reflected in Figure 1. We extracted mRNA expression of the oxytocin pathway in the Nucleus Accumbens in our secondary analysis of GTEx data, as this is one of the sampled brain region sites, but this was for the purposes of validating the observed patterns in the Allen data set. Thus, we did not make any inferences regarding a Nucleus Accumbens/OXT connection.

3. Minor point, but Figure 2 d-f would benefit from region labels. Relatedly, I am not sure that the current version of figure 2 is that meaningful. As noted in the figure, the hypothalamic expression of OXTR/CD38 are higher in hypothalamus compared to thalamus. Can the authors provide their rationale for choosing the thalamus as the control region? Figure 1 already shows that it has lower than average expression of OXT markers.

Response: We attempted adding labels for all regions for Figures d-f, however, we found that legibly-sized labels made the figures unclear. We provide full region labels in Supplementary Figure 7. We chose the thalamus as a control region given its close proximity to the hypothalamus and that data was sufficient data was available in the Allen dataset regarding gene expression in specific thalamic substructures. We provide this information in the manuscript:

“Given the role of the hypothalamus in oxytocin signaling, expression patterns within hypothalamic substructures were also summarized using anatomic labels from the Allen dataset (Fig. 2; Supplementary Data 1). Oxytocin pathway gene expression in nearby thalamic substructures using anatomic labels from the Allen dataset were also summarized for comparison”

4. In Fig 3, I may be misreading this and I appreciate the relative strength of the correlations in panels B/C, but don't these show that the DA/ACh gene sets do not demonstrate globally strong correlations to oxytocin markers? A genome-wide WGCNA approach may be more appropriate here, but would likely not show that these gene groups are highly related to one another.

Response: We agree that DA/ACh gene sets do not demonstrate globally strong correlations, and now note this in the paper as follows:

“Hierarchical clustering revealed that oxytocin pathway genes (OXTR, CD38) cluster with some elements of the dopaminergic (DRD2, DRD5, COMT) and cholinergic (CHRM4 and CHRM5) systems, but not all, indicating the relationship between oxytocin pathway genes and dopaminergic/cholinergic gene sets were not globally strong”

A genome-wide WGCNA approach might have been more appropriate here, but given the focus on the oxytocin pathway system, this would have been beyond the scope of the paper.

5. Figure 5 seems to be a bit redundant in making the point that OXTR/CD38 are correlated to each other. Perhaps shift it to the supplement?

Response: We agree, we now move this figure to the supplement (Supplementary Figure 10).

6. The authors note: "Notably, among the top ten most positive mRNA map correlations for OXTR were Neurotensin Receptor 2 (NTSR2; $r_s = 0.78$), Glutamate dehydrogenase 2 (GLUD2; $r_s = 0.78$), and Glutamate dehydrogenase 1 (GLUD1; $r_s = 0.77$), and the top ten most positive correlations with CD38 included NTSR2 ($r_s = 0.82$), C12orf39 (Spexin; $r_s = 0.81$), Phosphoserine Aminotransferase 1 (PSAT1; $r_s = 0.79$), and GLUD1 ($r_s = 0.78$)." It may help the readers parse the analyses if they provide some theoretical context detailing what makes these relations "notable."

Response: This is a good suggestion, we now provide more context detailing what makes these relations notable:

“Among the top ten most positive mRNA map correlations for OXTR were Neurotensin Receptor 2 (NTSR2; $r_s = 0.78$), Glutamate dehydrogenase 2 (GLUD2; $r_s = 0.78$), and Glutamate dehydrogenase 1 (GLUD1; $r_s = 0.77$), and the top ten most positive correlations with CD38 included NTSR2 ($r_s = 0.82$), C12orf39 (Spexin; $r_s = 0.81$), and GLUD1 ($r_s = 0.78$). Associations with these specific genes are notable given their reported role in metabolic regulation^{35, 36, 37, 38}. Additionally, the gene with

the 6th strongest association with CD38 was PSAT1 ($r_s = 0.79$), whose disruption has been linked to schizophrenia ³⁹.”

7. Can the authors include a brief note acknowledging that OXT transcripts are only expected to be present in OXT producing cells, which may be why it does not show whole-brain correspondence to OXTR/CD38. These data indicate the receptors are transcribed at site of uptake, but OXT is produced in known oxytonergic hypothalamic regions.

Response: We now provide a brief note acknowledging that OXT transcripts are only expected to be present in OXT producing cells, which may be why it does not show whole-brain correspondence to OXTR/CD38 expression:

“OXT expression did not demonstrate strong whole-brain correspondence with OXTR and CD38 expression, which was likely due to OXT transcripts only showing a strong presence in oxytocin producing cells within hypothalamic regions.”